# Investigations of Correlation and Coherence in Turbulence from a Large-Eddy Simulation

Regis Thedin, Eliot Quon, Matthew Churchfield, and Paul Veers

National Renewable Energy Laboratory

**Correspondence:** Regis Thedin (regis.thedin@nrel.gov)

**Abstract.** Microscale flow descriptions are often given in terms of mean quantities, turbulent kinetic energy, and/or stresses. Those metrics, while valuable, give limited information about turbulent eddies and coherent turbulent structures. This work investigates the structure of an atmospheric boundary layer using coherence and correlation in space and time with a range of separation distances. We calculate spatial correlations over entire planes of velocity fluctuations, from which we can evaluate the correlation along different directions at different spacings. Similarly, coherence of the three velocity components over separations in the three directions is also investigated. We apply these analyses to a mesoscale-to-microscale coupled scenario with time-varying conditions and examine nuances in spatial correlations that are often overlooked. Through these analyses and results, this work highlights important differences observed in terms of coherence when comparing large-eddy simulation data to simpler models, and suggests ways to improve these simpler models. We note that such differences are important for disciplines like wind energy structural dynamic analysis, in which blade loading and fatigue depend strongly on the structure of the turbulence. We emphasize the additional wealth of data that can be provided by typical atmospheric boundary layer large-eddy simulation when correlation and coherence analysis is included, and we also state the limitations of large-eddy simulation data, which inherently truncate the smaller scales of turbulence.

## 1 Introduction

Offshore deployment of wind farms offers a new set of challenges. As wind turbines increase in rotor diameter, it becomes increasingly important to characterize the flows these large turbines will experience. Knowing how the flow structures evolve over the increased geographic extent of wind plants is relevant to turbine and wind plant design. In fact, a better understanding of atmospheric and wind farm flow physics has been identified as one of the grand challenges in wind energy research (Veers et al., 2019), noting the coupling between mesoscale and microscale flows.

Descriptions of the microscale flow, or the turbulent flow of the atmospheric boundary layer, are usually given in terms of integral flow characteristics, such as statistics of mean and turbulent quantities. Such metrics, while valuable (e.g., see Robertson et al. (2018)), provide limited information about the spatial or temporal structure of turbulent eddies and how they change as background atmospheric conditions and stability change. Space–time correlation and coherence provide the necessary base for understanding the relation between spatial and temporal scales of motion. The correlations quantify how the fluctuations at one location relate to those at a different location, or how fluctuations at a point are related in time. Coherence

is similar, but compares the two points in the frequency domain rather than comparing the values in the time domain. Together, these quantities describe important characteristics of turbulent flow structure across spatial and temporal scales.

The turbulence models suggested by the international wind turbine design standards (IEC 61400-1, 2019) to predict mechanical loads contain mathematical descriptions of spatial coherence. One of the two models, which will be discussed in details in this work, includes exponential-based equations that impose coherence on a flow based on spectral content descriptions. Accurately capturing how coherences vary with different model parameters, such as separation distance, and change with different velocity components can better inform and improve models. The base of the suggested model go back to the original exponential decay coherence formulation of Davenport (1961), which states that the coherence spectrum $\gamma$ of the streamwise turbulent component $u$ is given by

$$\gamma = \exp\left(\frac{-C_z^u \delta_z f}{\overline{U}}\right), \tag{1}$$

where $C_z^u$ is an empirical decay coefficient, $\delta_z$ is the vertical separation distance, $f$ is the frequency, and $\overline{U}$ is the mean wind speed. Davenport's model, however, fails to account for the reduction of coherence at low frequencies and large separation distances. While in the original formulation the decay coefficient does not depend on the separation distance, it was later found that the dependency was necessary for vertical and lateral separations (Simiu and Scanlan, 1996; Saranyasoontorn et al., 2004; Bowen et al., 1983; Sacré and Delaunay, 1992; Cheynet et al., 2017b).

Davenport's model assumptions were also found to be invalid in situations where the separation distance is large with respect to the length scale of the turbulence (Kristensen and Jensen, 1979; Mann et al., 1991). Based on Davenport's model, Kaimal's spectrum with exponential coherence model (Kaimal et al., 1972; Thresher et al., 1981) addressed shortcomings related to the invalidity of Davenport's model in these situations. An additional term involves the ratio of the separation distance to a coherence scale parameter $L_c$. This extra term allowed for the reduction in coherence levels at zero frequency as the separation distance increases. The final equation will be given later and discussed in more detail. Such improvement eventually became one of the recommended models in the mentioned IEC guidelines (IEC 61400-1, 2019). The other model suggested in the standard is Mann's spectral turbulence model (Mann, 1994, 1998). The Mann model, on the other hand, is based on the von Kármán (1948) model, and assumes that an isotropic energy spectrum is distorted by a linearized mean velocity shear, providing one-point spectra, cross-spectra, and the coherence of the three components. In this work, we will focus on the Kaimal's model with the exponential decay from Davenport's model.

The use of models suggested by the IEC standards can result in an overestimation of fatigue loads (Holtslag et al., 2016). The same study, however, noted that the primary sources of fatigue loads, the wind shear and turbulence levels, can vary significantly depending on the stability state. While the IEC-suggested models were developed for neutrally stratified flows, such stability is often not the norm. The effect of stability on loads has also been studied in Sathe and Bierbooms (2007) and Sathe et al. (2013) by the use of Mann's model with the Monin–Obukhov length. While turbulence and shear, separately, cause different loads on blades, tower, and rotor, it was noted that the IEC standards are very conservative on the definition of wind shear and turbulence, which results in a significant overestimation of the loads (the authors note up to 96%) when compared

to loads obtained when using wind conditions specific to a site of interest. Note that the standard is of conservative nature by design.

Understanding how the coherence changes with varying conditions is important. Recent studies have assessed the effects of coherence functions on loads of offshore wind turbine blades (Doubrawa et al., 2019; Nybø et al., 2020). The impact of coherence functions has also been noted in other studies (Kelley et al., 2005; Kelley, 2011). Prior work focused on offshore flows, either by showing the coherence levels of such flows alone (Naito, 1983) or by establishing comparisons between observed data to Kaimal and Mann spectral models (Cheynet et al., 2017a). Often, all three components of the turbulence vector are analyzed, adding to the prior studies that only considered the streamwise component (e.g., Eidsvik (1985); Andersen and Løvseth (2006)). Unfortunately, usually only vertical separation is studied, given limited instruments arranged in a tower. Most of these studies found that coherence levels based on observations at an offshore environment are higher than those computed by the spectral models. Cheynet et al. (2018), however, found good agreement between exponential coherence models and observations under near-neutral conditions at sea for the streamwise component.

Coherence is also studied in the context of dynamic wake meandering. It has been reported that immersing wind turbines in a flowfield created by synthetic turbulence generators (e.g. TurbSim (Kelley and Jonkman, 2005)) can result in significantly different wake meandering behavior (Wise and Bachynski, 2019, 2020; Shaler et al., 2019). The differences in wake meandering appear when the flowfield is created by applying coherence only in the streamwise component, rather than in all components, when compared to wake characteristics obtained using turbulence-resolving large-eddy simulation tools. Some of these differences include negligible lateral wake meandering in cases without the application of lateral and vertical coherence, estimated from tracking the center of the wake laterally. As mentioned, the IEC standard only specifies in the coherence of the streamwise component in the vertical and lateral directions. The scales of interest for dynamic wake meandering are different that those scales that are known to impact loads and blade fatigue.

Obtaining the field data required to gain a comprehensive picture of the coherence and correlation of the flow can be challenging—especially with lateral coherence. Meteorological masts are typically deployed in isolation or with spacing that is much larger than the size of background turbulent structures, such that the fluctuations are decorrelated between the meteorological masts. It is worth noting, however, that instruments such as sonic anemometers and scanning Doppler lidars have been mounted on bridges in coastal areas, allowing a relatively successful study of lateral coherence (Kristensen and Jensen, 1979; Cheynet et al., 2016). Nonetheless, computational models, especially turbulence-resolving large-eddy simulations (LES), are particularly well-suited for the task because data can be collected anywhere in the flow field at high frequency. Prior work used LES (Simley et al., 2016; Berg et al., 2016; Lukassen et al., 2018; Doubrawa et al., 2019; Nybø et al., 2020) to assess coherence in the flow and compare with models, and in general it was found that the buoyancy effects affects shear and turbulence levels, which has a direct effect on the coherent structures. Both the Mann and the Kaimal with Davenport's exponential decay models, in their original form as suggested by the IEC standard, do not explicit account for potential temperature stratification and resulting buoyancy effects present in the atmosphere.

In this work we employ LES to compute the flow field within the atmospheric boundary layer and use the generated data to investigate the correlations and coherence present in all components of the turbulence and how they vary over time with varying

atmospheric conditions. In particular, the microscale simulations are performed with mesoscale forcing so that regional-scale

weather variations in the wind speed, direction, shear, and surface heat flux are introduced into the microscale domain. The goal is to highlight the additional information that can be obtained with LES, show how it relates to simple models, and note shortcomings that can benefit from further LES studies.

## 2   Methodology

### 2.1   Numerical Setup

The simulations are executed using the Simulator for Wind Farm Applications (SOWFA) (Churchfield et al., 2012), an LES code designed for atmospheric and wind energy applications. The simulation is done on a laterally periodic domain. The domain extends for 3 km in the horizontal directions and 1 km vertically, and has a uniform grid resolution of 10 m. The microscale mean profiles of velocity and potential temperature are driven toward mesoscale mean profiles as computed by the Weather Research and Forecasting (WRF) numerical weather prediction tool through a profile assimilation technique (Allaerts

et al., 2020). The conditions investigated are given in the next subsection. The code is executed with second-order-accurate schemes in space and time. The time step is chosen so that the Courant number does not exceed 0.75. A turbulence "spin-up" time is considered prior to the window of interest and ignored in the analysis. A differentiating aspect of this work is that the analysis is carried out on transient background conditions, driven by mesoscale mean quantities.

     We note that the domain extent implicitly limits the maximum correlation and coherence distance, as well as the maximum

integral length scale that the simulation is able to capture. However, investigations with larger domain sizes indicated that 3 km is well suited for the flow field observed during the period of interest.

### 2.2   Scenario Investigated

The focus of this work is on a 4-hour period in the North Sea off the German coast near the Netherlands at the FINO1 atmospheric measurement platform, see Fig. 1. The wind during the period of interest is predominantly from the northwest, so

from the open sea.

     The period of interest spans from 1 a.m. to 5 a.m. local time on May 16, 2010. The stability state over most of the period is slightly convective. We performed mesoscale-driven LES of the flow in the vicinity of FINO1 during the time period of interest. The overall background conditions are shown in Fig. 2. While we allow the conditions to change, the period of interest was picked because of the relative small change in wind direction and wind speed. The goal was to have the flow mostly from the

offshore environment, rather than influenced by the nearby land. High-frequency sonic anemometer data at the FINO1 tower were available at 40, 60, and 80 m above sea level (approximately). Unless otherwise noted, the comparisons with LES are made using observation data at 80 m above sea level.

     Turbulence intensity levels of the microscale LES followed well the levels observed in the FINO1 platform, shown in Fig. 2(c). A difference in both wind direction and wind speed can be observed in Figs. 2(a–b). The microscale does not receive

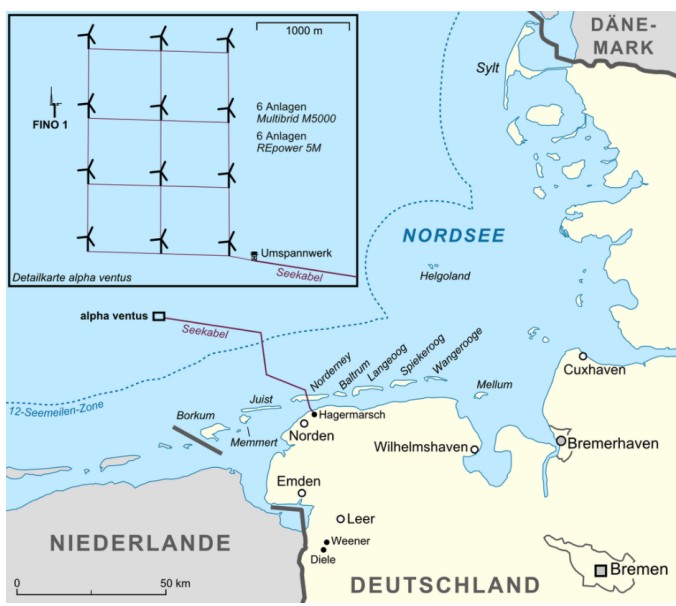

**Figure 1.** Overview of the region around the Alpha Ventus FINO1 tower. Figure from Wikimedia Commons, distributed under a CC BY-SA 3.0 license.

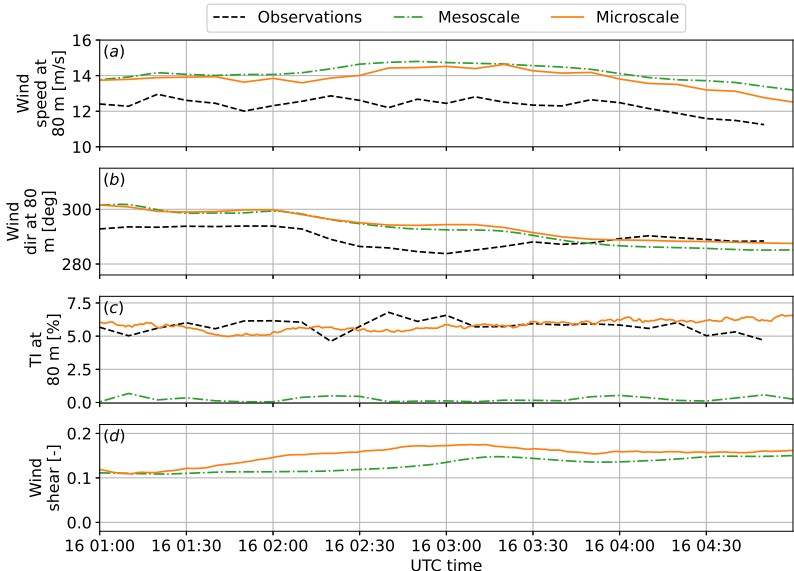

**Figure 2.** Mean background conditions during the period of interest compared to observation data and the mesoscale solution used to drive the microscale LES. (a) Wind speed, (b) wind direction, and (c) turbulence intensity at 80 m, (d) wind shear exponent.

any information from the observations, but rather from the mesoscale, which means the microscale will inherit any errors from the mesoscale and the microscale mean quantities will only be as accurate as the mesoscale. Note that turbulence is not resolved at mesoscale resolutions, reflected in Fig. 2(c). The mesoscale coupling provides information on mean profiles, resulting in similar shear histories. Shear information is not explicitly passed from the mesoscale to the microscale. The shear exponent value in Fig. 2(d) was calculated using a logarithmic curve-fit on the bottom 500 m of the boundary layer. Nonetheless,

by allowing the shear levels to fluctuate from that implicitly given by the mesoscale, the microscale was able to capture the appropriate level of turbulence of that captured by instruments at sea.

The velocity spectra, Fig. 3, show the differences between the small-scale turbulence resolved by LES and that measured at the FINO1 for the first hour and the last hour of the period of interest. There is a good match between LES and observation data before the drop-off in resolved content by LES. The drop-off in energy for the LES occurs between 0.1 and 0.2 Hz, a result of

its inability to capture such frequencies due to grid size limitations and mean wind speeds over the interval. The energy content of the LES is slightly higher than the content of the observation data set for the last hour of the period at the low-frequency range, possibly a result of a delay of the LES turbulence to react to the slight ramp-down event that started at 04Z (Fig. 2(a), thus still exhibiting higher levels of turbulence intensity (Fig. 2(c)). The velocity spectra are obtained using Welch's algorithm with an overlapping (50%) 15-min Hamming window. No significant differences were observed by using a Hann window.

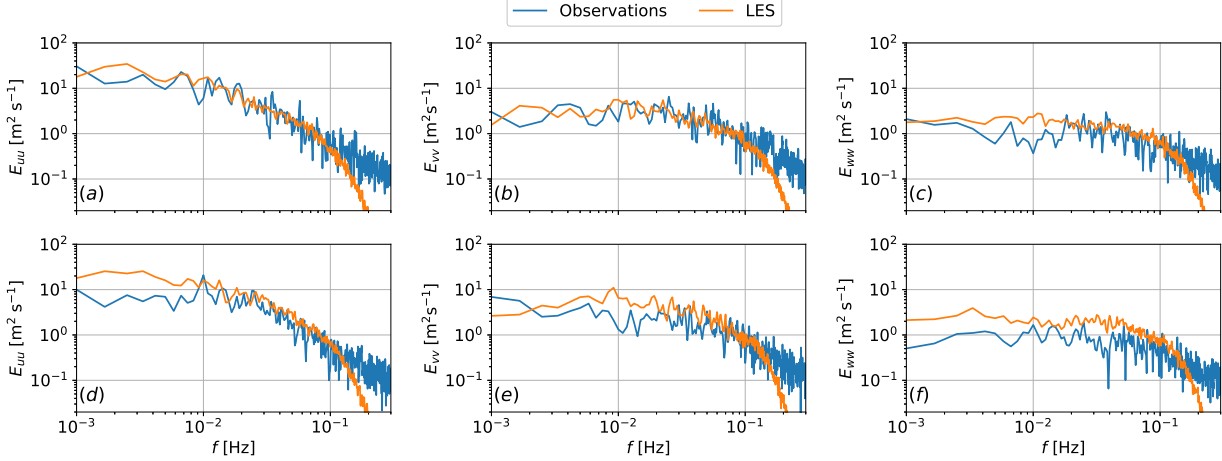

**Figure 3.** Comparison of power spectra of the three turbulence components (a,d) $u$, (b,e) $v$, and (c,f) $w$ for the first hour (top row), and the last hour (bottom row) of the period of interest at 80 m using 15-min Hamming windows.

## 2.3 Turbulence Spatial Correlation

The correlation coefficient, $R_{ij}$, between two points $\mathbf{x}$ and $\mathbf{x} + \mathbf{r}$, where $\mathbf{r}$ is a separation vector, is given by

$$R_{ij}(\mathbf{x},\mathbf{r},t) = \frac{\langle u_i(\mathbf{x}+\mathbf{r},t)\,u_j(\mathbf{x},t)\rangle}{\sqrt{\langle u_i^2(\mathbf{x}+\mathbf{r},t)\rangle}\sqrt{\langle u_j^2(\mathbf{x},t)\rangle}} \tag{2}$$

where $u_i$ denotes the zero-mean turbulent fluctuations, and the angled brackets denote an ensemble average of realizations. Here we focus on autocorrelations (i.e., $i = j$). Throughout this work, we define $u = u_1$ as the streamwise component, $v = u_2$ as the cross-stream component, and $w = u_3$ as the vertical component of the flow. For clarity, when speaking of velocity components, we refer to "streamwise" and "cross-stream" components. However, when speaking of the directionality of separation vector, we refer to "along-wind" and "crosswind" directions.

The idea is to perform two-point correlation computations with respect to a fixed point. We vary the second point so that the turbulence over a plane covering the computational domain is correlated with the fixed point. The result is a map of the correlation coefficients over this plane. This is useful in assessing how the turbulence evolves and allows us to obtain correlation coefficients between two points arbitrarily spaced. For a horizontal plane at reference height (e.g., 80 m), we start by saving horizontal slices at 1 Hz from the LES for postprocessing. Picking the central point as the reference point ($\mathbf{r} = 0$), we expect (by definition) the correlation to be exactly 1 at this central point with an exponential decay with increasing $\mathbf{r}$. The domain size allows us to perform correlations with $|\mathbf{r}| \leq 1.5$ km.

The procedure is to apply Eq. (2) to all points on the plane with respect to a central point for each snapshot. Time averaging replaces ensemble averaging, but because the mesoscale conditions vary in time, a temporal average over the whole interval is not performed. Instead, shorter time averaging windows are used over which mesoscale conditions change relatively little. From a study on window size and overlap, we found that a 15-min window with 10-min overlap provided (i) enough data for smooth converged statistics, (ii) a short enough interval such that mean conditions did not change appreciably, and (iii) a large enough interval to accommodate large time-scale features. A goal of this analysis is to see how the correlation coefficients change with evolving conditions, and not necessarily to capture short, instantaneous transients.

It is important to highlight the assumptions and limitations so far. In choosing the grid resolution, a bound on the smallest resolved scales is imposed, which for this type of numerical method is 4–5 times the grid resolution (Pope, 2001, p. 574). In choosing a domain extent, a limit on the largest scales is also imposed. Domain sizes, though, are usually much larger than the largest scales of interest. We assume horizontal homogeneity due to the choice of periodic lateral boundaries on the case setup.

## 2.4 Turbulence Coherence over Arbitrary Separations

Knowing the integral length scales present in the flow, we compute a related but different statistic, coherence, with separation distances of the same order of magnitude. Here we focus on coherence magnitude, which we will simply refer to as "coherence." It is the normalization of the magnitude of the cross spectra of velocity fluctuations. In other words, it describes the correlation between two time series as a function of frequency, but does not give information about the phase. The square of coherence magnitude between two signals $i$ and $j$ is defined as

$$\gamma_{ij}^2(f) = \frac{|S_{ij}(f)|^2}{S_{ii}(f) \, S_{jj}(f)} \tag{3}$$

where $S_{ii}$ and $S_{jj}$ are the power spectral density of signals $i$ and $j$, and $S_{ij}$ is the cross-power spectral density between $i$ and $j$. The two signals are of individual components of the three-dimensional velocity vector $\mathcal{U} = (u, v, w)$ and are often given in terms of along-wind, crosswind, or vertical separation distance $\delta$. The type of coherence is given by the direction of the

separation. For instance, the along-wind coherence of the streamwise component is given by $\gamma^2_{uu,\text{long}}$, where the two time series are of streamwise velocity at two locations separated by distance $\delta$ in the along-wind direction.

Coherence and correlation give similar information, but they differ in a few ways. Correlation shows how two quantities are related in physical space or time and can vary from $-1$ to $1$. For example, if two sinusoidal signals are identical in amplitude and frequency, but have a phase lag at a certain separation distance, the correlation can lie anywhere between $-1$ and $1$ depending on the phasing. For instance, for a separation that aligns maxima with minima, the correlation will become anti-correlated with a value of $-1$. On the other hand, coherence magnitude ranges from $0$ to $1$ and is not as sensitive to the phase lag because it uses cross-spectra magnitude. Two identical but phase-lagged sinusoidal signals will have a coherence of $1$ at only the frequency of the sine wave (the other frequencies would be undefined due to zero spectral information and the definition (3)). Although this work focuses on coherence magnitude, one can also examine the real and imaginary part of the coherence, which are termed "co-coherence" and "quad-coherence," respectively. While coherence magnitude is a measure of the consistency of a phase relationship in the data, it does not have information on what that phase relationship is—such information can be obtained by individual examination of co-coherence and quad-coherence separately.

The Kaimal spectrum model with exponential decay coherence model, suggested by the IEC standards, is only defined in the crosswind and vertical directions. An improvement to Davenport's exponential model given in Eq. 1, the Kaimal's model introduces an additional term that is a function of the separation distance, $\delta$, and a coherence scale parameter, $L_c$. For the streamwise component of the velocity at two points, vertically or laterally separated by distance $\delta$, with mean wind speed $\overline{U}$, the coherence model reads

$$\gamma^2_{uu_{\text{lat,vert}}}(f,\delta) = \exp\left(-a\sqrt{\left(\frac{f\,\delta}{\overline{U}}\right)^2 + \left(\frac{b\,\delta}{L_c}\right)^2}\right) \tag{4}$$

where $a$ and $b$ are tuning parameters. The IEC standards recommend $a = 12$ and $b = 0.12$. $L_c$ is given as $8.1\Lambda_1$, where $\Lambda_1$ is a longitudinal scale parameter constant at 42 m for hub heights above 60 m.

Another common model for coherence adopted by the IEC guidelines is the Mann (1998) model. Both the Kaimal and Mann models are based on spectral methods. While the Mann model is based more on physics and is a function of parameters used to define the spectral tensor, the Kaimal model is based more on empirical formulations and less on physics. While they are computationally inexpensive and useful in the design process, they have two main limitations. First, in some cases, when the flow field as computed by these models is used to drive load calculations, the results can be different from one model to another (Eliassen and Obhrai, 2016), resulting in inconsistencies in load estimations. A second limitation of the models is that atmospheric stability is not considered, as neutral stratification provided a sufficient description of the turbulence for load estimation purposes at the time the models were developed.

To add longitudinal separation and to overcome the atmospheric stability limitation, studies have proposed more complex models for longitudinal coherence. For example, adding longitudinal separation, Simley and Pao (2015) suggested

$$\gamma^2_{uu_{\text{long}}}(f,\delta) = \exp\left(-\left(a_1\frac{\sigma}{\overline{U}} + a_2\right)\sqrt{\left(\frac{f\,\delta}{\overline{U}}\right)^2 + \left(\frac{b_1\,\delta}{L_c^{b_2}}\right)^2}\right) \tag{5}$$

where $\sigma$ is the standard deviation of the wind speed, $L_c$ is a measure of the integral length scale, and $a_1$, $a_2$, $b_1$, and $b_2$ are empirical constants adjusted using LES. In general, a number of exponential forms of the correlation decay have been proposed. For more information, the interested reader should see the review by Martin et al. (2015).

As mentioned, an accurate modeling of coherence is an important task. An increase in coherence has been shown to increase the loads (Eliassen et al., 2015). The study of wind-induced response of wind turbines can be traced back to Davenport (1962) with the development of the buffeting theory, which allowed coherence and one-point velocity spectra to be used to predict the dynamic response of wind-sensitive slender structures. Nowadays, more complex simulation tools for wind energy applications such as load estimation and thus design of wind turbines often follow IEC standards, which suggest Eq. (4) for the streamwise velocity, thus neglecting the other components as well as the longitudinal separations. Prior work focused on observation data has investigated other components in other separations (Saranyasoontorn et al., 2004).

The estimation procedure for coherence and some results are presented in section 3.2. As shown earlier in Fig. 3, the highest frequency properly resolved by the LES given the grid resolution is approximately $10^{-1}$ Hz. Due to this recognized inability of our current setup to capture higher frequency phenomena, our focus will be at the low-frequency range of $f < 0.15$ Hz, which corresponds to approximately the rotational frequency of large offshore wind turbines.

## 3 Results

### 3.1 Correlation Results

Performing the steps outlined in section 2.3 on the whole domain results in few realizations and noisy results. Keeping the largest scales of interest in mind, we leverage horizontal homogeneity and use spatial averaging as ensemble averages. Thus we split the domain into smaller subdomains and perform local spatial correlation analysis. For this subdomain analysis, the correlation between all locations within a sub-domain and its central point is established. Figure 4 shows the correlations for the three components of turbulence on a 3-by-3 grid of $1 \times 1$ km subdomains. This strategy allows for more averaging and smoother statistics; however, it imposes a tighter limit on the largest scales. The aforementioned temporal windowing imposes another limit on the largest time scale captured. The limit on the time scale is not relevant for this problem, but it is worth mentioning. Figure 4 shows an average over the whole 4-hour period of interest—the intent of this figure is to illustrate the process and not directly obtain data from it. The general wind direction (roughly southeast, see Fig. 2) can be qualitatively observed in the correlation of the streamwise component.

The spatial average of the subdomains shown in Fig. 4 can be obtained for each 15-min window separately. A single interval is shown next in Fig. 5 for illustration purposes. Note the axis limits and overall domain size after the ensemble average. The figure includes arrows indicating the mean wind direction over the interval. From this point forward, all results are related to the correlation analysis performed using the 3-by-3 grid of $1 \times 1$ km subdomains.

The results of this analysis show what is often intuitive by looking at typical boundary layer flow fields: the turbulent structure of the streamwise component of the turbulence is "stretched" in the along-wind direction, whereas structures formed by the other components are much more isotropic. The behavior of the streamwise component is much different in the crosswind

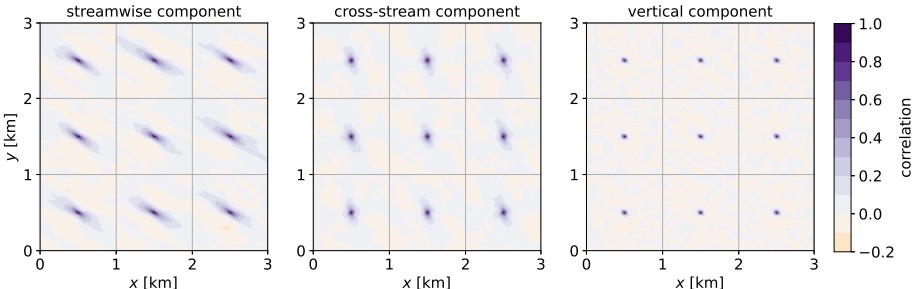

**Figure 4.** Contour plots of spatial correlation for the three turbulence components for the split-domain approach. Nine domains organized in a 3-by-3 grid are use. The goal is to have more realizations for an ensemble average. For each panel, a spatial average is shown in Fig.5. The general wind direction, towards the southeast, can be observed in the streamwise component.

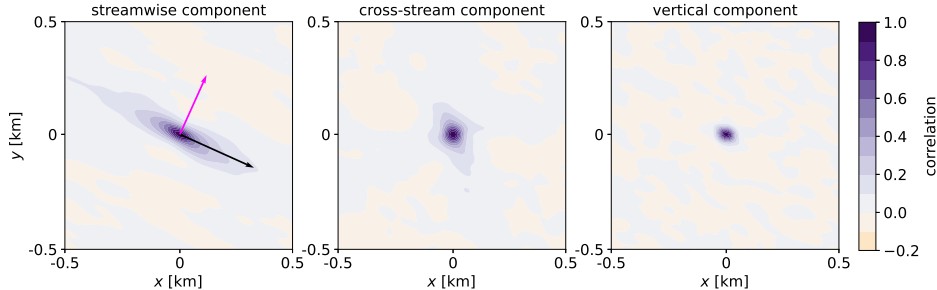

**Figure 5.** Contour plots of spatio-temporal average of spatial correlation for the three turbulence components for the 15-min interval starting at 2:40Z. The interval-mean along-wind and crosswind directions are indicated by the black and magenta arrows, respectively. The spatial correlation of the streamwise component is stretched in the direction of the wind (mostly towards the southeast, black arrow), while other components show no clear preferential direction.

versus the along-wind direction: within a relatively short distance, it becomes decorrelated and then slightly anticorrelated, meaning that one should expect alternating patterns of along-wind elongated structures containing streamwise velocity excesses arranged next to deficits. This spatio-temporal correlation analysis show similar results to that of Lukassen et al. (2018).

Because we compute autocorrelation maps for each overlapping time window within the analysis period, we can observe
how turbulence autocorrelation varies as flow conditions change. Although the contour maps are very informative, we sought ways to reduce the information they contain to quantities of interest that we can track versus time. As a first step, for each 1-Hz snapshot of each 15-min interval, we sample autocorrelation coefficient values from the contour maps over along-wind- and crosswind-oriented lines that pass through the central point (see black and magenta arrows in Fig. 5, which indicate the direction of these lines). These lines are constant for each interval and represents the mean wind direction within that interval.
The resulting curves for each snapshot are averaged in time and represents the correlation coefficient in the along-wind and crosswind direction of that interval. The resulting curves of correlation coefficients versus along-wind or crosswind distance are

shown in Fig. 6 for the streamwise component of the flow. In this figure, each light blue curve comes from the ensemble average at each time interval of 15 min. The red curve is the average of all the individual 15-min curves. The red curve is computed in order to establish a more direct comparison with observation data over the entire period of interest, as will be discussed later.

By definition in Eq. (2), the correlation coefficients are 1 at zero separation. The more gradual decay of correlation coefficient with along-wind versus crosswind distance is clear. The autocorrelation of streamwise velocity fluctuation drops to effectively zero within 150 m in the cross-stream direction. However, in the along-wind direction, the correlation coefficient decay to zero is not fully captured over the half-length of the subdomain, even though the decorrelation length scale can be seen to be around 400–450 m for some of the time intervals.

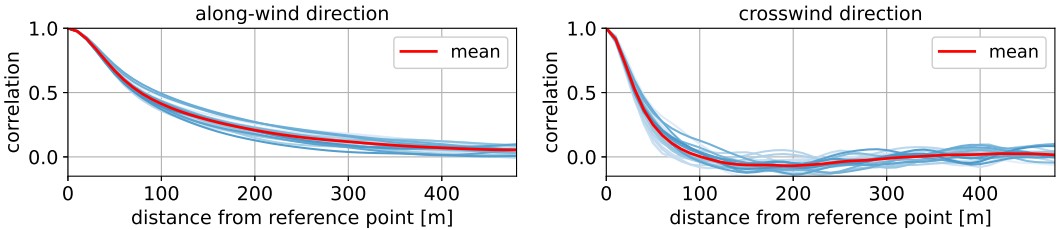

**Figure 6.** Correlation coefficients from Fig. 5 in the along-wind and crosswind directions for the streamwise component. Each blue curve represents a 15-min interval, whereas the red curve is the mean over the time period of interest.

In the above spatial autocorrelation analysis, we examined spatial correlations where the separation vector is purely horizontal. The FINO1 data investigated come from a single meteorological mast that only contains vertical spacings between measurement points. To this end, we compute temporal autocorrelations of the streamwise component of the observation data in order to compare it to the spatial correlations, using

$$R_{ij}(\mathbf{x},\tau) = \frac{\langle u_i(\mathbf{x},t_0+\tau)\,u_j(\mathbf{x},t_0)\rangle}{\sqrt{\langle u_i^2(\mathbf{x},t_0+\tau)\rangle}\sqrt{\langle u_j^2(\mathbf{x},t_0)\rangle}} \tag{6}$$

where $\tau$ is a separation in time. We transform between spatial separation and temporal separation in the along-wind direction by assuming the turbulence is frozen and advecting with the mean wind (Taylor, 1938).

To test the frozen turbulence assumption more definitively, we compute and compare the temporal correlation on the field and LES data. The results are shown in Fig. 7. Figure 7(a) shows temporal autocorrelations of the streamwise component of the velocity of the observed data, obtained at 80 m. In Fig. 7(a–c), each light-shade curve represents the correlations of a

270 15-min interval, whereas the dark shade curves are the average of the light curves, representing the average of the full 4-hour period of interest. Note that in Fig. 7(a), the individual curves are rather noisy. Therefore, to obtain some sort of ensemble average, we perform an average over the whole period. This average, indicated by the dark shade, can be compared with LES results. In Fig. 7(b), we show the temporal autocorrelations of the streamwise velocity components collected from nine virtual meteorological masts in the LES, so it is directly analogous to the observation results from Fig. 7(a). In Fig. 7(c), we show the

275 correlation presented prior in Fig. 6 along the mean wind direction, converted to the time domain using Taylor's hypothesis.

Because of the spatial subdomain size and the mean advection speed, the maximum time separation computed is about 36 s. The reach of this curve is a direct result of splitting the domain. Finally, Fig. 7(d) shows the mean results (dark shade) of each Fig. 7(a–c) together for ease of comparison. Based on this exercise, noting the good match between the red and green curves, we conclude that frozen turbulence appears to be a reasonable assumptions and that it can be used to transform between spatial and temporal correlations. We note a mismatch between curves based on observed data versus LES in low values of time separation (from 0 to 10 s). This mismatch is possibly caused by the inability of the LES to resolve turbulence below its spatial and temporal filter scale.

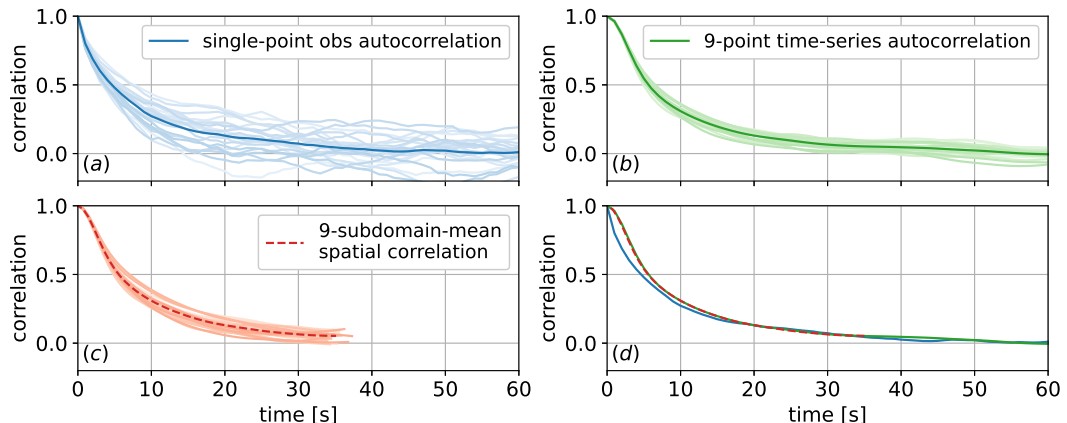

**Figure 7.** Comparison of spatial correlation and temporal autocorrelation of streamwise component of the velocity from LES with field data. Each lightly shaded curve represents one 15-min interval; the dark-shaded curve is the average of the light curves. (a) Temporal autocorrelation of single-mast observation data obtained at 80 m; (b) Temporal autocorrelation of 9 virtual meteorological masts from LES; (c) Spatial correlation results from LES transformed to time-domain; (d) Comparison of the mean curves of each panel.

The integral length scales of the streamwise component in the along-wind direction $L_u^x$ and crosswind direction $L_u^y$ are obtained by integration of the spatial correlation curves presented in Fig. 6. The results are shown in Fig. 8(a), where integration is carried out until the correlation drops to 0.05 (value suggested by Flay and Stevenson (1988), and used by others such as Tian et al. (2018)). The same process is performed with the integration of the temporal autocorrelation curves to obtain the integral time scale. Again, we may use Taylor's assumption to transform between the integral time and length scales, which was used to produce Fig. 8(b). In canonical LES studies, stationary mesoscale conditions are usually the focus. Here, with mesoscale coupling and varying mesoscale conditions, we are able to study the effect of mesoscale transients on turbulence. In the conditions investigated in this work and as shown in Fig. 8, the integral length scale in the along-wind direction $L_u^x$ mostly fluctuates between 110 m to 150 m. In comparison, the integral length scale of the streamwise component in the crosswind direction $L_u^y$ is about 3 to 4 times smaller that in the along-wind direction.

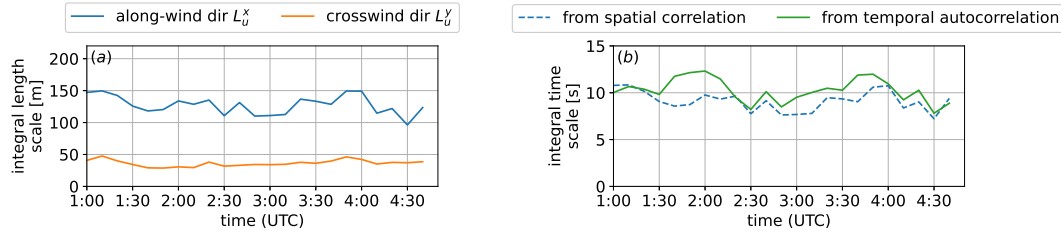

**Figure 8.** Integral scales variation of the 4-hour period of interest from LES. (a) Integral length scale of the streamwise component of the turbulence $u$ for the along-wind direction ($L_u^x$), and crosswind direction ($L_u^y$); (b) Along-wind integral time scale calculated separately from spatial correlation and temporal autocorrelation data.

## 3.2 Coherence Results

For vertically separated pairs of turbulence fluctuations, we sampled a vertical plane along the domain. For longitudinal and lateral separations, we use the same horizontal plane as the prior analysis. Interpolation is not needed for the vertically separated coherence, but it is needed within grid points for longitudinal and lateral separation. It is challenging to have grid points that are aligned with the wind direction because of the changing wind conditions.

The time series during the period of interest is not stationary, thus the spectra and subsequently the coherences are calculated using smaller intervals. The sample results shown in this section were obtained with 1 hour of data, using 50%-overlapping 15-min windows multiplied by the Hann function.

Figure 9 shows the curves obtained when focusing on the low-frequency range for the first hour of the interval of interest. There is no recommendation by the IEC standards for longitudinally separated points, so comparisons with the Simley and Pao (2015) model are presented for the streamwise velocity component, where it is defined.

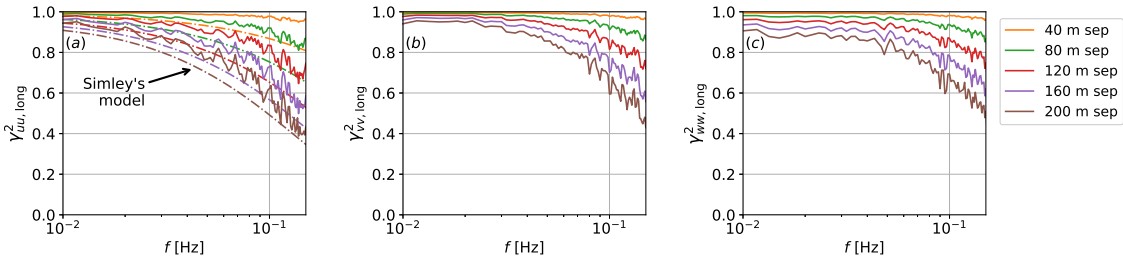

**Figure 9.** Coherence of the three turbulence components (a) $u$, (b) $v$, and (c) $w$ separated longitudinally. Also shown are Simley's model, Eq. (5). Note the logarithmic scale of the $x$-axis.

All three components show a high value of coherence at a relatively wide range of frequencies. The results match well the stability state correction proposed by Simley and Pao (2015). The results shown for the streamwise component of longitudinally

separated points are also similar to those obtained using lidar data (Debnath et al., 2020), where for a case with about 140-m length scale, the drop-off occurred at around 0.1–0.2 Hz. No model is available for the other components.

Coherence of longitudinally separated points is not as important as laterally and vertically separated points from a wind turbine design point of view. While the high levels of coherence in the longitudinal direction is a well-known fact, it is still nonetheless relevant the quantification of the longitudinal coherence. Performing this analysis on longitudinal separation is relevant in the context of wind turbine control, where controls strategies based on inflow preview require knowledge of the evolution of the turbulence in the longitudinal direction, as it approaches the rotor. Some studies have looked exclusively at longitudinal coherence with controls, rather than loads, in mind (see Schlipf et al. (2013, 2015), as well as the aforementioned Simley and Pao (2015); Debnath et al. (2020)).

Coherence of the components of the turbulence separated laterally are shown in Fig. 10 for selected separation distances. Where Kaimal's model with Davenport's exponential decay is defined, comparisons are provided. The IEC-recommended model overestimates the coherence in the frequency range investigated. The coherence values for laterally separated points (Fig. 10) are significantly lower than those encountered in longitudinally separated points (Fig. 9), as expected. The lateral, as well as vertical, coherence may impact turbine loads, especially as wind turbine rotors become larger. For lower lateral separation values, the coherence drops at very low frequencies. In fact, it has been long known that the coherence does not approach 1 as frequencies approach zero (Kristensen and Jensen, 1979; Saranyasoontorn et al., 2004). The results are consistent with those found by Bardal and Sætran (2016) where it was noted that given the same separation, lateral coherence is "significantly" smaller than longitudinal coherence. At higher frequencies, the computed curves do not converge to zero, which stems from the fact that the coherence definition (3) used here is a biased estimator (Kristensen and Kirkegaard, 1986; Mann, 1994), although the lack of convergence towards zero has also been attributed to numerical noise in the work of Shaler et al. (2019).

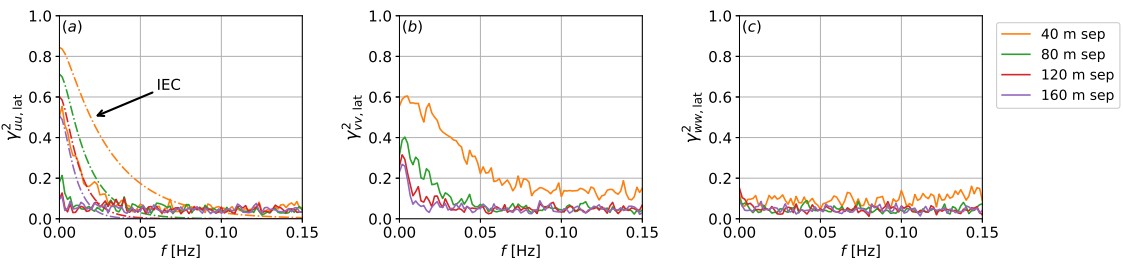

**Figure 10.** Coherence of the three turbulence components (a) $u$, (b) $v$, and (c) $w$ separated laterally. Dash-dotted curves shown are obtained from the IEC-recommended Kaimal spectrum with exponential coherence model, where defined.

The coherence for separations in the vertical direction is shown next in Fig. 11, with comparisons with Kaimal's spectrum with Davenport's exponential coherence model. In this case, the separation value is given relative to a measurement point at 80 m. We acknowledge that 80 m is too low for typical and next-generation offshore wind turbine hub heights—this reference point was chosen due to the data availability from the FINO1 research platform equipment. We use the term "negative separation" to indicate vertical separation directionality. Although using a negative separation distance in a coherence model would be

nonphysical, we emphasize that here negative separation means that one measurement point is below the reference point located at 80 m. The other curves presented with positive separation means they were computed with the pair of points consisting of the reference point at 80 m and the second point being above the reference by the separation distance. The plots show a faster decay in the coherence of the streamwise component than in the other components, as well as a slight overprediction by Kaimal's spectrum with exponential coherence model. The vertical component decays the slowest with vertical separation. The results are consistent with prior investigations by Saranyasoontorn et al. (2004) on much smaller separation distances. A small asymmetry between positive and negative separations is observed, and can be a result of the fact that the largest eddies present increase with height. Interestingly, for the vertical component, the negative separations result in stronger coherence decay with increasing frequency than that for the corresponding positive separations. Comparing the same separation magnitudes (the pairs of separations 20 and −20, and 40 and −40), we note an asymmetry effect that appears to increase as the separation distance gets larger. This asymmetry effect has been first noted and modeled in the work of Bowen et al. (1983), and further investigated and improved in Cheynet (2018), which essentially takes the two heights into account, rather than just a separation distance, while also accounting for the non-unity at zero frequency. We note that curves related to 20-m separation distance are showed for illustration and should not be deemed well-resolved, as such separation includes only two LES grid points, and because of that, we isolate the curves related to a separation of 40 m in Fig. 12 for ease of comparison. Finally, the vertical coherence of $v$ and $w$ components do not approach 1 as the frequency tends to zero. In similar observations, Naito (1983) attributed the cause to be due to the fact that that these components rarely include long-period fluctuations in the surface layer. Nonetheless, as mentioned before, Kristensen and Jensen (1979) point out that the coherence is not unity because the separation distance is not negligible when compared to typical length scales of turbulence.

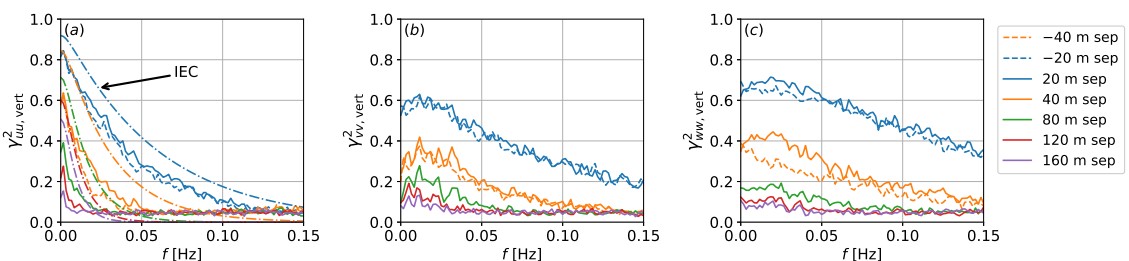

**Figure 11.** Coherence of the three turbulence components (a) $u$, (b) $v$, and (c) $w$ separated vertically with respect to 80 m. Negative separation means that one of the points is below 80 m. Note the asymmetry shown by the same separation in different directions. Dash-dotted curves shown are obtained from the IEC-recommended Kaimal spectrum with exponential coherence model, where defined.

One interesting aspect of the IEC standard when it comes to recommendations of the Kaimal spectrum with exponential coherence model is that no distinction is made between lateral and vertical coherences. This is an important aspect that is not often investigated by the literature. We show in Fig. 12 only the curves related to a separation of 40 m in both the lateral and vertical direction, alongside the Kaimal spectrum with exponential coherence model prediction for the same separation. The intention here is to highlight that the levels of coherence can be substantially different depending on the direction and

355 component of the flow. As we have previously mentioned, coherences in the $v$ and $w$ components are needed for realistic wake meandering predictions.

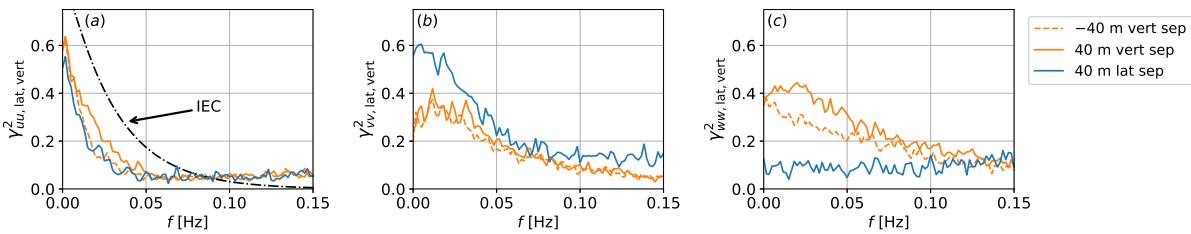

**Figure 12.** Comparison of lateral and vertical coherences related to separations of 40 m. Negative separation means one point is below the reference point located at 80 m, while positive separation means the second point is above the reference point.

To assess if the LES results are within what one would expect, we compare two of the vertical separation curves to the observed data. The observation data were available at three heights, thus allowing two different separation distances with respect to the reference height of 80 m to be analyzed. Figure 13 shows the result. Note that prior comment about 20-m

separation also applies here, as it constitutes only two LES grid points and thus results encompass a greater uncertainty. For the streamwise component there is a good match, with the observation data following the LES much better than Kaimal's model. The rather noisy nature of the observations curve is due to single-mast data. For the cross-stream and vertical components, the general trends and decay rate are also captured. We point out, nonetheless, that a model based on LES data could provide more information than no model at all. The present study suggests that, if considering vertical coherence over large separation

distances, it might be important to consider the aforementioned asymmetry effect. With large separation distances, relevant for tall, large offshore wind turbines, the overall distance to the sea (or ground) level can lead to different characteristics in coherence (as shown in Fig. 11(c)). For example, the coherence between points at 40 m and 120 m above sea level is unlikely to be the same as the coherence between points located at 120 m and 200 m above sea level, even though they are separated by the same distance.

An advantage of using large-eddy simulation to obtain coherence is that we are able to perform the same analysis for an arbitrary separation distance. Some curves obtained at discrete separation values have been presented, but in Fig. 14 we show a contour plot of the same quantities for all separation distances. Although this figure also shows very small separations, we remind the reader that the values in such regions are not well resolved because the finest computational grid resolution is 10 m. The goal here is to point out an advantage of numerical models over field observations in that data can be sampled at virtually

any location. Without that ability, we could not create the complete maps of coherence and correlations shown in Figs. 5 and 14.

Figure 14 summarizes the importance of modeling all three components of the turbulence. Perhaps not surprisingly, the highest coherence occurs for the component of turbulence in the same direction as the separation direction (diagonal panels). The coherence of large separation distances do not approach 1 as the frequency tends to zero, as reported in the literature

(e.g., Doubrawa et al. (2019)).

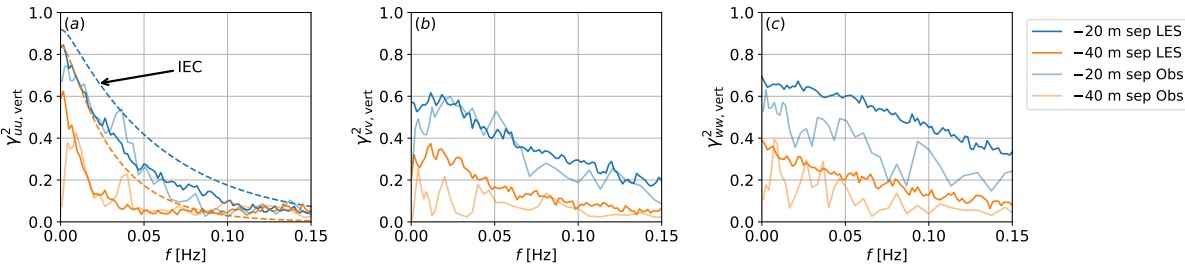

**Figure 13.** Comparison of coherence related to observed data for (a) $u$, (b) $v$, and (c) $w$ components of the wind speed separated vertically with respect to 80 m. Negative separation means one point is below the reference point located at 80 m, while positive separation means the second point is above the reference point. Dash-dotted curves shown are obtained from the IEC-recommended Kaimal spectrum with exponential coherence model, where defined.

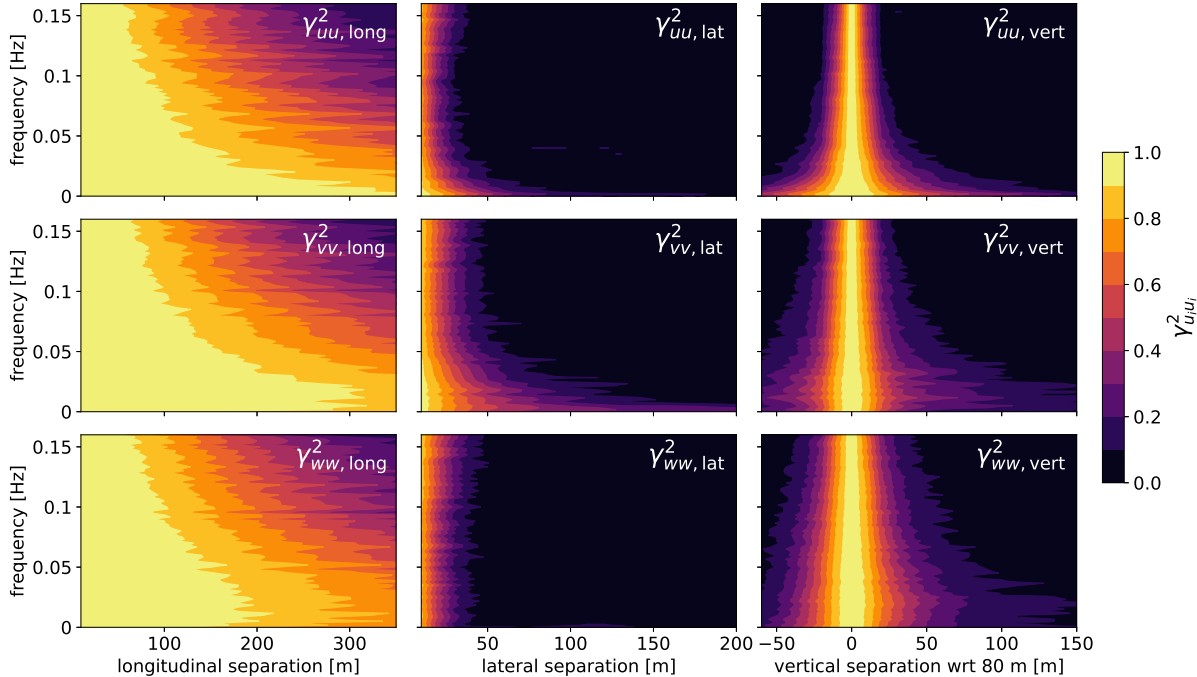

**Figure 14.** Contour of the coherence for all components of the turbulence, along all three separation directions, over a continuous range of separation distances. The diagonal panes are the component in the same direction of the separation.

## 3.3 Cross Coherence

We calculate cross coherences $uv$, $uw$, and $vw$, each obtained from two distinct components of the turbulence at zero separation (same point). For the calculation of cross coherences shown here, we select points that lie on the 80 m above-ground-level plane

and perform the computation. For each point, cross coherences are calculated and then an ensemble average is computed using all the individual cross coherences. The results for the first hour of the interval is are shown in Fig. 15.

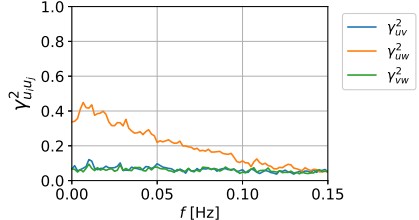

**Figure 15.** Mean cross coherence for points at a plane 80 m above ground level.

Cross coherence is not usually considered in preliminary design and load analysis of new wind turbines. The isotropic von Kármán model (von Kármán, 1948) considers all of them to be zero, due to the isotropy assumptions and lack of shear consideration. The Mann uniform shear model considers only $uw$ coherence to be nonzero. This is what we observe in the cross coherence shown in Fig. 15. The largest cross coherence is found between the streamwise and vertical components, while no significant correlation exists between the streamwise and cross-stream turbulent components, and neither between the cross-stream and vertical components (ignoring numerical noise). These findings are consistent with those from Saranyasoontorn et al. (2004) and the aforementioned study by Mann (1994). The $uw$ coherence is related to the friction velocity and, therefore, affected by the wind shear. This study adds to the study of Saranyasoontorn et al. (2004) pointing out that it might be relevant to account for the effect of $uw$-coherence when generating inflow with the goal of load analysis.

## 4 Discussion

The analysis procedure outlined in this study is general and can be applied to any LES solution of the atmospheric boundary layer. The curves obtained through the analysis can then be used to inform a synthetic turbulence generator such as Turb-Sim (Kelley and Jonkman, 2005). TurbSim has the option to generate synthetic inflow following the Kaimal spectrum model with exponential decay coherence, options that may be readily extended to use coherence characteristics obtained from a high-fidelity solution. Future work should further study the variation of coherence curves of each component for any arbitrary separation distance given varying atmospheric conditions, which will produce a set of coherence curves for each component and separation direction. The additional curves can be used to derive curves for intermediate conditions that were not explicitly simulated. Ultimately, this would enable LES-informed time series of turbulence to be generated for any condition using synthetic turbulence generators. Prior studies such as Simley and Pao (2015) looked at canonical stable and unstable conditions in an LES setting, while others such as Cheynet (2018) (and references therein) looked at continuous data for an offshore case, but only for a limited set of separation distances and for the streamwise component. More accurate time series of the turbulence can improve the representation of design load conditions that are used for load estimations in lower-fidelity wind turbine design codes. The ultimate impact is the ability to obtain less conservative, site-specific designs, relying less on simplified models like

that of Kaimal with Davenport exponential decay, one of the suggested models the current IEC standards is based on. It is also worth noting that wind turbine design is not the only discipline that can benefit from more accurate coherence estimations. Longitudinal and lateral coherence are relevant in the control of wind turbines and wind farms. Longitudinal coherence affects feed-forward control (e.g., Simley and Pao (2015); Schlipf et al. (2013); Debnath et al. (2020)), while lateral coherence is important for dynamic wake meandering studies and can affect the efficacy of wake tracking and wake steering (e.g., Wise and Bachynski (2020); Shaler et al. (2019)).

The coherence magnitude is given by the cross-spectra normalized by the auto spectra of the individual components of the turbulence. The coupled effects of the different components of the turbulence are expressed in the cross-spectra term. From a statistical perspective, the information contained in the coherence fully defines two-point, second-order statistics of random fields. Therefore, all the second-order dynamic properties are contained in these quantities and can be used to generate a field that is consistent with those properties. We emphasize that these are the quantities that dominate the dynamic response of a system like a wind turbine rotor.

The asymmetry aspect inherent to the study of vertical coherence is captured in this work, adding to the body of literature. In close relationship with the prior points raised in this section, the variation of such asymmetry effect given different stability states, and how that further affects the loads and fatigue characteristics, is an important question. While out of scope of the current analysis, we note that the mesoscale-coupled LES setups presented here are more realistic than the standard "canonical" setups and and enable the investigation of coherence effects in site and weather-condition specific situations.

It is also important to note the grid spacing used in this work and the final applicability of such a model on modern large, flexible rotors. The 10-m resolution limits the scale of the resolved turbulence and frequency of the computed coherence. For instance, resolving turbulent structures at 20-m separation can be relevant near the tip of the blade and, at the current grid spacing, 20-m separation results are not reliable. The general analysis applied to simulations with higher spatial resolution would allow much more detailed investigations of the coherence in all three components, informing both how we model and measure turbulence. This realization certainly drives the authors' decision making about LES resolution in our future work.

A higher spatial resolution would also push the cutoff frequency higher than the 0.15 Hz shown here and provide insight into higher-frequency, smaller-scale turbulence. The procedure is also suitable for modeling turbine–turbine interactions in a wind farm setting. For example, considering a row of turbines, one could process the data following the coherence analysis presented here and come up with simple analytical models that could potentially include the wake of turbines within the farm. Ultimately, such models can be used within other simulation tools for loads and fatigue. Some of these other simulation tools are lower in fidelity than an LES approach and runs much faster, allowing efficient and robust iterative design of wind turbines. Some of these tools include OpenFAST and FAST.Farm (Jonkman et al., 2017). Flow fields generated using stochastic tools with LES-informed coherence characteristics can be further used for wind farm analysis in, e.g., FAST.Farm, which would ultimately result in better estimates of loads and wind-turbine array efficiency.

Another interesting application of the analysis routine outlined is its use in the assessment of whether or not the resolved turbulence has reached a "fully developed" stage. Consider the common situation in which a wind farm LES is set up with inflow without grid-scale-resolved turbulence. For example, some researchers apply inflow to their LES that comes from a

mesoscale weather model which does not explicitly resolver turbulence. Such an LES setup exhibits a *fetch* region in which both resolved and modeled turbulent quantities must undergo a transient to a fully-developed state. The length of this spatial fetch region is often determined by visual inspection of the flow field and, more quantitatively, using power spectral density. The correlation analysis of streamwise and lateral flow presented here can be used as another metric to quantify when a fully-developed state has been reached and to help determine the fetch region extent.

## 5 Concluding Remarks

This work highlights the utility of computing turbulence correlations and coherence using turbulence-resolving LES data. Coherence of the $v$ and $w$ components along all three directions does not appear to be negligible, but the Kaimal model with Davenport's exponential coherence—the basis for one of the models suggested by the IEC standards—is only defined for the $u$ component. This missing information that is essential to creating coherent synthetic turbulence may be filled in by knowledge gained through LES.

We showed that evolving conditions can change the way the flow is presented in terms of both integral scales and coherence levels. For the period investigated, when the wind shear remained relatively constant with an exponent of 0.1–0.15, our analysis indicates that frozen turbulence appears to be a suitable assumption when correlations are the main metric under investigation. The coherence decay, however, is faster in some components and directions than others. This suggests that given high coherence for eddies of certain sizes, the turbulence can be considered frozen; while the same may not be true for eddy sizes corresponding to low coherence values. The variation of coherence decay among the different directions and components of the wind speed can be significant. This is an important aspect of some flows that may be overlooked by simplified studies where single-point first-order metrics (e.g., turbulent kinetic energy) are highlighted. A better understanding of the spatial structure of the turbulence under different conditions can improve turbulence models that are used for loads calculations. A higher confidence in the estimations from such models could result in better estimates of blade fatigue life. We also note that a more accurate representation of coherence characteristics are useful for other areas such as the modeling of wake meandering and inflow-preview-based control strategies.

This work demonstrates the wealth of additional information that can be gathered from a typical atmospheric LES, and outlines the numerical limitations that need to be considered when using LES data. State-of-the-art atmospheric simulation capabilities are continuing to evolve, and complex, large scenarios are now more routine practice. Some of these complex scenarios include realistic weather conditions such as full diurnal cycles, frontal passages, and low-level jets, among others. With the rapid adoption of GPU-based LES codes, we are headed into ever faster, higher-resolution workflows. Quantities such as correlation, integral scales, coherence, and cross-spectral information are not often computed and discussed in typical atmospheric and wind plant studies. We therefore have an opportunity to develop new insights from site- and condition-specific studies without any additional computational expense, and to advance the state of the science in characterizing the atmospheric boundary layer and modeling turbulence.

*Code and data availability.* The LES code SOWFA is available at github.com/NREL/SOWFA-6. Input files for the case analyzed in this work are available at github.com/a2e-mmc/SOWFA-setups/tree/master/offshore_FINO1_intCoupled. Output data are available upon request.

*Author contributions.* This work was led by RT, who performed the simulations and carried out the analysis. RT and MC formulated the research goals and aim of the study. MC, EQ, and PV supervised the conceptualization, and contributed to discussions on the formal analysis.
RT prepared the manuscript with contributions and review from all co-authors.

*Competing interests.* One of the co-authors is a member of the editorial board of Wind Energy Science.

*Acknowledgements.* This work was authored by the National Renewable Energy Laboratory, operated by Alliance for Sustainable Energy, LLC, for the U.S. Department of Energy (DOE) under Contract No. DE-AC36-08GO28308. Funding provided by the U.S. Department of Energy Office of Energy Efficiency and Renewable Energy Wind Energy Technologies Office. This research was performed using compu-
485 tational resources sponsored by the Department of Energy's Office of Energy Efficiency and Renewable Energy and located at the National Renewable Energy Laboratory. The views expressed in the article do not necessarily represent the views of the DOE or the U.S. Government. The U.S. Government retains and the publisher, by accepting the article for publication, acknowledges that the U.S. Government retains a nonexclusive, paid-up, irrevocable, worldwide license to publish or reproduce the published form of this work, or allow others to do so, for U.S. Government purposes. The authors thank Patrick Hawbecker from NCAR for providing the mesoscale dataset used in this work. FINO1
data made freely available by the Bundesamt für Seeschifffahrt und Hydrographie (BSH) agency is greatly appreciated. The authors thank Etienne Cheynet and other anonymous reviewer for their thorough review and comments provided.

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
