# Peer review of "Investigations of Correlation and Coherence in Turbulence from a Large-Eddy Simulation"

_Wind Energy Science, 2022_

## Referee Comment (RC1)

**Investigations of Correlation and Coherence in Turbulence from a Large-Eddy Simulation**

Reviewer's comments

**General comment**

The manuscript "Investigations of Correlation and Coherence in Turbulence from a Large-Eddy Simulation" by Thedin et al deals with the study of 2-point characteristics of turbulence in the marine atmospheric boundary layer (MABL) by the mean of Large Eddy Simulation (LES). These characteristics are the auto-correlation function, integral time and length scales and coherence. The paper is of broad international interest and reads well. The topic is within the scope of Wind Energy Science. The manuscript discusses many relevant scientific questions but addresses them superficially only. Despite the potential of the numerical tools displayed by the authors, I am afraid that the analysis is currently too shallow to be considered for publication in Wind Energy Science. Maybe focusing on one or two specific questions would lead to more significant conclusions.

**The fundamental importance of the literature review**

The manuscript needs to include a more attentive literature review on the coherence of turbulence to identify which aspects of the coherence of natural wind are needed in wind energy. The coherence of turbulence has been studied during the 1970s and 1980s by e.g. Pielke and Panofsky (1970); Ropelewski et al. (1973); Kristensen and Jensen (1979); Kristensen et al. (1981); Bowen et al. (1983). I am aware that these studies are fairly old. Yet, they are still highly relevant to the present work and actually answer some of the questions raised by the authors. Similarly, the work by Davenport (1962) and other wind engineers in the 1960s and 1970s (e.g. Scanlan, 1978) is central to connecting the concept of coherence and wind loading on the structure. I understand well that reviewing the past literature is tedious. Nevertheless, it is a crucial task to better anchor the present study into contemporary challenges of wind loading on wind turbines.

**Identifying a more specific research question**

Besides the literature review, the choice of research question should prioritize the important challenges for wind turbine design. This was partly advertised in the abstract, which I found well written and successfully caught my attention. Yet, it seems that these challenges were not clearly identified in the present manuscript. In the following, I attempt to name a few of them. I hope these can be useful to the authors:

- The study of the lateral coherence of turbulence. Focusing on the coherence at lateral separation and discussing how it relates to the coherence at vertical separations is highly

valuable. Indeed, there are only a few studies focusing on the lateral coherence in the MABL and those relying on lidar instruments (e.g. Cheynet et al., 2016, 2021) still face considerable challenges. As correctly pointed out by the authors, meteorological masts are rarely deployed in arrays. Therefore, large Eddies Simulation (LES) offers an excellent occasion to study the lateral coherence. In particular, it would be valuable to know whether the lateral coherence can be inferred from the vertical coherence. This can be discussed with respect to the uniform shear model (Mann, 1994) and/or in terms of decay coefficient from the Davenport model (see Solari and Piccardo (2001) for a review of these coefficients).

- The authors point out that the asymmetry effect, related to the blocking by the ground and the non-linear variation of the Davenport decay coefficient with the separation distance (e.g. Bowen et al., 1983; Sacré and Delaunay, 1992). The role of the blocking by the ground on the structure of turbulence is mentioned in Mann (1994). Accounting for the asymmetry effect may be significant for wind turbine design (Cheynet, 2019). The dependency of the asymmetry effect on atmospheric stability is not well known. Studying this dependency and quantifying it through LES simulation is a fascinating topic that could be a stand-alone paper.

- There exist only a few studies of the coherence of turbulence above the atmospheric surface layer (ASL) (e.g. Lothon et al., 2006). A large portion of the rotor plane of modern offshore wind turbines is located above the ASL. How to scale the coherence of turbulence above the ASL? Can the depth of the atmospheric boundary layer influence the coherence above the ASL? LES could give some insight into these questions. This topic is maybe more related to atmospheric science, but it could still be relevant to wind energy science.

**Specific comments**

Additional specific comments were summarized as "Community Comment" in Cheynet (2022).

**References**

Pielke, R., Panofsky, H.. Turbulence characteristics along several towers. Boundary-Layer Meteorology 1970;1(2):115–130.

Ropelewski, C.F., Tennekes, H., Panofsky, H.A.. Horizontal coherence of wind fluctuations. Boundary-Layer Meteorology 1973;5(3):353–363.

Kristensen, L., Jensen, N.. Lateral coherence in isotropic turbulence and in the natural wind. Boundary-Layer Meteorology 1979;17(3):353–373.

Kristensen, L., Panofsky, H.A., Smith, S.D.. Lateral coherence of longitudinal wind components in strong winds. Boundary-Layer Meteorology 1981;21(2):199–205.

Bowen, A.J., Flay, R.G.J., Panofsky, H.A.. Vertical coherence and phase delay between wind components in strong winds below 20 m. Boundary-Layer Meteorology 1983;26(4):313–324.

Davenport, A.G.. The response of slender, line-like structures to a gusty wind. Proceedings of the Institution of Civil Engineers 1962;23(3):389–408.

Scanlan, R.. The action of flexible bridges under wind, II: Buffeting theory. Journal of Sound and vibration 1978;60(2):201–211.

Cheynet, E., Jakobsen, J.B., Svardal, B., Reuder, J., Kumer, V.. Wind coherence measurement by a single pulsed Doppler wind lidar. Energy Procedia 2016;94:462–477.

Cheynet, E., Flügge, M., Reuder, J., Jakobsen, J.B., Heggelund, Y., Svardal, B., et al. The cotur project: Remote sensing of offshore turbulence for wind energy application. Atmospheric Measurement Techniques Discussions 2021;2021:1–32. doi:10.5194/amt-2020-511.

Mann, J.. The spatial structure of neutral atmospheric surface-layer turbulence. Journal of fluid mechanics 1994;273:141–168.

Solari, G., Piccardo, G.. Probabilistic 3-D turbulence modeling for gust buffeting of structures. Probabilistic Engineering Mechanics 2001;16(1):73–86.

Sacré, C., Delaunay, D.. Structure spatiale de la turbulence au cours de vents forts sur differents sites. Journal of Wind Engineering and Industrial Aerodynamics 1992;41(1-3):295–303.

Cheynet, E.. Influence of the measurement height on the vertical coherence of natural wind. In: Conference of the Italian Association for Wind Engineering. 2019, p. 207–221.

Lothon, M., Lenschow, D.H., Mayor, S.D.. Coherence and scale of vertical velocity in the convective boundary layer from a Doppler lidar. Boundary-layer meteorology 2006;121(3):521–536.

Cheynet, E.. Comment on wes-2022-71. 2022. doi:10.5194/wes-2022-71-CC1.

---

## Community Comment (CC1)

**Investigations of Correlation and Coherence in Turbulence from a Large-Eddy Simulation**

**Community comment by Etienne Cheynet**

The manuscript "Investigations of Correlation and Coherence in Turbulence from a Large-Eddy Simulation" by Thedin et al. addresses major contemporary challenges for the development of wind energy. I found the preprint informative and interesting to read. Nevertheless, the manuscript has some shortcomings. A number of points should be clarified, partly because the scientific review of the coherence and its connexion to wind loading is perfectible.

**1 Specific comments**

**Point 1**

Abstract: I feel the abstract is somewhat self-contradictory. The authors write first that turbulence statistics and spectral analysis give limited information to describe a turbulent flow. Then the authors state that they focus on the coherence and spatial correlation functions. But the coherence is also a (two-point) spectral characteristic of turbulence. The integral time scale and integral length scale are also turbulence characteristics[a]. So I do not really understand the logical reasoning here.
* * *
[a]The turbulence length scales are integral characteristics and contain less information than the coherence or auto- and cross-covariance functions

**Point 2**

Line 23-25: It should be noted that the coherence does not necessarily compare the phase relationship between two time series or spatial series. In eq. 3, the authors use the magnitude-squared coherence (mscoh) which does not give any information on the phase. This is something they judiciously comment. A more appropriate definition of coherence is given by Ropelewski et al. (1973), which states that the coherence is a correlation function in the frequency space.

**Point 3**

Line 31-32: I understand what the authors mean with "implicitly", but this sentence is actually incorrect. In the Davenport model (Davenport, 1962), the separation distance is explicitly needed

(and modelled) in the coherence function:

$$\gamma_u = \exp\left(\frac{-C_z^u d_z f}{\bar{u}}\right) \tag{1}$$

where $d_z$ is the vertical separation distance, $f$ is the frequency, $C_z^u$ is an empirical decay coefficient and $\bar{u}$ is the mean wind speed. Interestingly, in the Davenport model, the dependency of $C_z^u$ on the separation distance is not modelled. This dependency was found to be, sometimes, necessary for vertical separations (Bowen et al., 1983; Cheynet, 2019) and lateral spatial separations (Sacré and Delaunay, 1992; Cheynet et al., 2017).

**Point 4**

Line 33: Contrary to a popular belief, Kaimal et al. (1972) did not study the coherence of turbulence. Therefore, writing "Kaimal's coherence" is incorrect. A more appropriate wording for this coherence model would be "exponential decay model" based on the Davenport model. It can also be noted that the empirical velocity spectra model used in IEC 61400-1 (2005) is not Kaimal's model either, although the work by Kaimal et al. (1972) was used as a basis for this model.

**Point 5**

Line 41: We should note that in the marine atmospheric boundary layer (MABL), non-neutral conditions are also commonly observed at wind speed above the rated wind speed of wind turbines (Barthelmie, 1999; Sathe et al., 2011; Cheynet et al., 2018).

**Point 6**

Line 53-54: The sentence "In general, these studies found that coherence levels based on observations at an offshore environment are higher than those computed by the spectral model" may be contradicted by the scientific literature. In Cheynet et al. (2018), the IEC exponential coherence model is found to model fairly well to the vertical coherence under near-neutral conditions at FINO1. However, as for the Davenport model, the IEC exponential coherence model has two main drawbacks: (1) it does not account for the thermal stratification of the atmosphere and (2) its main decay coefficient, which is equal to 12, is independent on the separation distance. In Mann (1994) and Cheynet (2019), the uniform-shear (US) model tends to slightly overestimate the vertical coherence of turbulence. It can be noted that, following Mann (1994), the US model captures remarkably well the lateral coherence of near-neutral turbulence.

**Point 7**

Page 2: I agree with the authors that studying the lateral coherence with sonic anemometers is challenging. However, anemometers mounted on bridges in coastal areas (e.g. Sacré and Delaunay, 1992; Kristensen and Jensen, 1979) have been used in the past with encouraging results. For the past few years, scanning Doppler wind lidars have also been used to study the lateral coherence

of turbulence (with relative success). Sonic anemometer data from bridges, while not offshore, are still valuable to validate spectral turbulence models. Also, the US model Mann (1994) was validated for lateral separation in the MABL. This may be worth mentioning in the manuscript.

**Point 8**

Line 162-163: A more robust way to test Taylor's assumption of frozen turbulence is to study the coherence of turbulence with longitudinal separations. If the turbulent field is frozen, the longitudinal coherence will be equal to 1 at every frequency. In reality, it will not be the case. This method can also be used to show that smaller eddies will not satisfy the assumption of frozen turbulence at separation greater than a few dozen of meters while bigger eddies may be considered as "frozen" over a distance greater than e.g. 100 m (Cheynet et al., 2017, Fig. 11). The approach adopted in Fig 6 is more limited as it only focuses on large eddies.

**Point 9**

The paper seems to dedicate a significant part to the coherence at longitudinal separation. From the viewpoint of wind turbine design, this may be a minor challenge compared to the need to reduce the uncertainties associated with the co-coherence at lateral and vertical separations.

**Point 10**

Line 240: The relationship between wind loading and wind coherence was developed and modelled in the 1960s by pioneers such as Davenport (Davenport, 1962), through the so-called "buffeting theory". This theory was originally applied to high-rise buildings and bridges. Maybe a lesser-known fact is that the response of a wind turbine can be quantitatively described without an elaborated numerical model by the buffeting theory. I recommend reformulating line 162 by stating that the theory related to the wind-induced response of wind turbines can be traced back to the 1960s with the development of the buffeting theory, where both the coherence and one-point velocity spectra were used to predict the dynamic response of wind-sensitive slender structures.

**Point 11**

Line 263-265 and Lines 270-271: It should be kept in mind that one fundamental assumption to study and model the coherence of turbulence is that the flow is fairly homogeneous and stationary. Downstream of a wind turbine and inside a wind farm, the flow can be strongly heterogeneous. Therefore, the paragraph on lines 263-265 and lines 270-271 may be criticized for violating the assumption of flow homogeneity. I think these lines can be removed without problem.

**Point 12**

Line 268-270: I do not understand the logical reasoning behind the sentence "The coherence values for laterally separated points [...] suggest that it may be important to properly account for the longitudinal separation, in addition to the lateral and vertical separations included in the

IEC standard". The fact that the lateral coherence $coh_y$ is smaller than the longitudinal coherence $coh_x$ does not justify the need to model $coh_x$ for wind loading on a wind turbine. If turbulence is considered frozen, which is commonly done, then for a given separation, $coh_x \geq coh_y$ at every frequency.

**Point 13**

Line 272-273: The fact that the coherence is not always equal to 1 at zero frequency was documented by Kristensen and Jensen (1979) and in many studies since then. This should be mentioned in the manuscript

**Point 14**

Line 274-276: The fact that the mscoh does not converge toward zero is not due to numerical noise but because the mscoh is a biased estimator. This bias is documented in e.g. Kristensen and Kirkegaard (1986) and commented in Mann (1994, p. 156). To avoid (or limit) this bias, I recommend focusing on the co-coherence.

**Point 15**

It should be noted that experimental studies of the coherence at separation distance greater than 100 m are associated with significant uncertainties because only a few data points are different from zero or are hidden in the ambient noise. To reduce such uncertainties, a large number of samples should be used. The coherence estimates will be then ensemble averaged. Ideally, both short and large separation distances should also be considered to capture the full extent of the coherence. If such conditions are not met, it can be challenging to reach meaningful conclusions.

**Point 16**

Page 14: I do not recommend using negative separations because these have no physical meaning and could trigger numerical errors with exponential coherence models. Instead, I suggest focusing on pairs of measurement heights or lateral positions as formulated in Putri et al. (2022, p. 1697).

**Point 17**

Line 280-288: From my understanding, the asymmetry effect mentioned by the authors has been studied in detail in the MABL by Cheynet (2019) at FINO1 and by Putri et al. (2022) near Vindeby wind farm. The asymmetry can be modelled quite well using the approach adopted by Bowen et al. (1983) and reflects the presence of the ground, which blocks the flow. Some interesting questions are (1) how this asymmetry changes with the thermal stratification of the atmosphere and (2) how it affects the loading on a wind turbine. These challenges can be discussed by the authors or be the topic of further study.

**Point 18**

Line 288-289: The justification with reference to Naito (1983) seems incorrect to me. The coherence is lower than unity at zero frequency because the separation distance is not negligible compared to a typical length scale of turbulence (Kristensen and Jensen, 1979). If there exist long-period fluctuations, then the time series may be non-stationary and prevent the study of turbulence characteristics. Fortunately, a quick look at Fig 2 suggests that the time series are reasonably stationary.

**Point 19**

Line 332-334: The suggestion "Future work should include a parametric study with varying atmospheric conditions, which will produce a set of coherence curves for each component and separation direction" has been done for vertical separations in Cheynet et al. (2018) at FINO1 using two years of continuous measurements. More generally, the dependency of the coherence on the atmospheric stability is also documented for an onshore location in Soucy et al. (1982) and in the MABL by Putri et al. (2022).

**Point 20**

Line 339-344: I agree with the authors. However, this paragraph is partly covered by the buffeting theory that was developed in 1960s and 1970s. So I am not sure if it is still an up-to-date material for discussion. Maybe this can be elaborated or removed?

**Point 21**

Section 5: One of the limiting aspects of the coherence for stochastic flow simulation is the need for a fairly homogeneous flow, which is not the case inside a wind farm. This limit can be partly overcome by LES simulation. maybe this aspect can be discussed in more detail here?

**References**

Ropelewski, C.F., Tennekes, H., Panofsky, H.A.. Horizontal coherence of wind fluctuations. Boundary-Layer Meteorology 1973;5(3):353–363.

Davenport, A.G.. The response of slender, line-like structures to a gusty wind. Proceedings of the Institution of Civil Engineers 1962;23(3):389–408.

Bowen, A.J., Flay, R.G.J., Panofsky, H.A.. Vertical coherence and phase delay between wind components in strong winds below 20 m. Boundary-Layer Meteorology 1983;26(4):313–324.

Cheynet, E.. Influence of the measurement height on the vertical coherence of natural wind. In: Conference of the Italian Association for Wind Engineering. 2019, p. 207–221.

Sacré, C., Delaunay, D.. Structure spatiale de la turbulence au cours de vents forts sur differents sites. Journal of Wind Engineering and Industrial Aerodynamics 1992;41(1-3):295–303.

Cheynet, E., Jakobsen, J.B., Snæbjörnsson, J., Mann, J., Courtney, M., Lea, G., et al. Measurements of surface-layer turbulence in a wide Norwegian fjord using synchronized long-range Doppler wind LiDARs. Remote Sensing 2017;9(10):977.

Kaimal, J.C., Wyngaard, J.C.J., Izumi, Y., Coté, O.R.. Spectral characteristics of surface-layer turbulence. Quarterly Journal of the Royal Meteorological Society 1972;98(417):563–589.

IEC 61400-1, . Iec 61400-3 wind turbines part 1: Design requirements. 2005.

Barthelmie, R.J.. The effects of atmospheric stability on coastal wind climates. Meteorological Applications: A journal of forecasting, practical applications, training techniques and modelling 1999;6(1):39–47.

Sathe, A., Gryning, S.E., Peña, A.. Comparison of the atmospheric stability and wind profiles at two wind farm sites over a long marine fetch in the North Sea. Wind Energy 2011;14(6):767–780.

Cheynet, E., Jakobsen, J., Reuder, J.. Velocity spectra and coherence estimates in the marine atmospheric boundary layer. Boundary-Layer Meteorology 2018;169(3):429–460.

Mann, J.. The spatial structure of neutral atmospheric surface-layer turbulence. Journal of fluid mechanics 1994;273:141–168.

Kristensen, L., Jensen, N.. Lateral coherence in isotropic turbulence and in the natural wind. Boundary-Layer Meteorology 1979;17(3):353–373.

Kristensen, L., Kirkegaard, P.. Sampling problems with spectral coherence; vol. 526. Risø National Laboratory Roskilde, Denmark; 1986.

Putri, R.M., Cheynet, E., Obhrai, C., Jakobsen, J.B.. Turbulence in a coastal environment: the case of Vindeby. Wind Energy Science 2022;7(4):1693–1710.

Soucy, R., Woodward, R., Panofsky, H.. Vertical cross-spectra of horizontal velocity components at the Boulder observatory. Boundary-Layer Meteorology 1982;24(1):57–66.

---

## Author Comment (AC2)

**WES 2022–71: Author's Response to the Reviewers**

The authors thank both reviewers for their time and valuable comments provided, which certainly helped improve the manuscript. Please find below specific responses to each comment.

**Reviewer 1**, Etienne Cheynet**

The authors thank Dr. Cheynet for his expertise and time spent in reviewing our manuscript, and providing such detailed and thorough comments. His detailed comments, submitted as a Community Comment, were certainly helpful in improving the manuscript. We have taken all the recommendations and have modified the article in every instance to meet the reviewer's requirements and suggestions. Individual replies to all points raised are given below. A marked-up version of the article highlights all the removed and added text.

**Reviewer Point P1.1** — The manuscript "Investigations of Correlation and Coherence in Turbulence from a Large-Eddy Simulation" by Thedin et al. deals with the study of 2-point characteristics of turbulence in the marine atmospheric boundary layer (MABL) by the mean of Large Eddy Simulation (LES). These characteristics are the auto-correlation function, integral time and length scales and coherence. The paper is of broad international interest and reads well. The topic is within the scope of Wind Energy Science. The manuscript discusses many relevant scientific questions but addresses them superficially only. Despite the potential of the numerical tools displayed by the authors, I am afraid that the analysis is currently too shallow to be considered for publication in Wind Energy Science. Maybe focusing on one or two specific questions would lead to more significant conclusions.

**Reply**: We appreciate the recognition of the importance of the topic and the scientific questions we have raised and discussed in the paper. We understand the reviewer's comment that the paper may be too shallow, and in some ways that may be true. It was not our goal to delve into any individual aspect of a particular turbulent flow under a particular set of conditions; instead, it was our goal to demonstrate the techniques and approaches that will allow other wind-energy researchers to exploit the power of large-eddy simulation in ways that it rarely is exploited. Most commonly researchers do not exploit large-eddy simulation data to explore turbulence structure information that is so crucial to wind-turbine response. We understand this is not a complete assessment of all the aspects of, for example, lateral or vertical coherence that are present in the LES data—that is a beyond the scope of a single paper. Here we demonstrate how a more complete assessment can be done, show what the results look like, apply the methods to advanced LES that is more realistic than standard because of the mesoscale weather coupling, and encourage other researchers to pursue this path—that is the goal of this paper. With this paper, we also aim to add to the literature on the uses of high-fidelity modeling tools to compute quantities that can help further inform the workhorse lower fidelity engineering models. Our discussion has revolved around the fact that extra information is available from these expensive numerical models and not many researchers are making use of them. We aimed to show how these tools compare with one of the simple models suggested by the IEC standard and highlight the potential for improvement of the simple models. These models have been around for decades and were developed when expensive simulation models were not feasible. Nowadays, the computational power allows us to tackle such problems. Making this clear was a goal with this manuscript. The authors believe that this paper is an important contribution to the literature — in addition to complementing several existing experimental analyses, we make a strong case for the use of high-fidelity modeling to improve simpler models typically used in turbine design. This work is also novel in the sense that it combines mesoscale coupling techniques in an offshore environment and coherence chacterization, which, to the authors' knowledge, has never been investigated concurrently.

**Reviewer Point P 1.2** — The manuscript needs to include a more attentive literature review on the coherence of turbulence to identify which aspects of the coherence of natural wind are needed in wind energy. The coherence of turbulence has been studied during the 1970s and 1980s by e.g. Pielke and Panofsky (1970); Ropelewski et al. (1973); Kristensen and Jensen (1979); Kristensen et al. (1981); Bowen et al. (1983). I am aware that these studies are fairly old. Yet, they are still highly relevant to the present work and actually answer some of the questions raised by the authors. Similarly, the work by Davenport (1962) and other wind engineers in the 1960s and 1970s (e.g. Scanlan, 1978) is central to connecting the concept of coherence and wind loading on the structure. I understand well that reviewing the past literature is tedious. Nevertheless, it is a crucial task to better anchor the present study into contemporary challenges of wind loading on wind turbines.

**Reply**: Thanks for pointing us to much of the important literature that we failed to mention in the paper. We have expanded the literature review and introduction part of the manuscript, accounting for most of the work the reviewer mentioned. We have also added several references to relevant work throughout the discussion of our results.

**Reviewer Point P 1.3** — Besides the literature review, the choice of research question should prioritize the important challenges for wind turbine design. This was partly advertised in the abstract, which I found well written and successfully caught my attention. Yet, it seems that these challenges were not clearly identified in the present manuscript. In the following, I attempt to name a few of them. I hope these can be useful to the authors:

- The study of the lateral coherence of turbulence. Focusing on the coherence at lateral separation and discussing how it relates to the coherence at vertical separations is highly valuable. Indeed, there are only a few studies focusing on the lateral coherence in the MABL and those relying on lidar instruments (e.g. Cheynet et al., 2016, 2021) still face considerable challenges. As correctly pointed out by the authors, meteorological masts are rarely deployed in arrays. Therefore, Large Eddy Simulation (LES) offers an excellent occasion to study the lateral coherence. In particular, it would be valuable to know whether the lateral coherence can be inferred from the vertical coherence. This can be discussed with respect to the uniform shear model (Mann, 1994) and/or in terms of decay coefficient from the Davenport model (see Solari and Piccardo (2001) for a review of these coefficients).
- The authors point out that the asymmetry effect, related to the blocking by the ground and the non-linear variation of the Davenport decay coefficient with the separation distance (e.g. Bowen et al., 1983; Sacré and Delaunay, 1992). The role of the blocking by the ground on the structure of turbulence is mentioned in Mann (1994). Accounting for the asymmetry effect may be significant for wind turbine design (Cheynet, 2019). The dependency of the asymmetry effect on atmospheric stability is not well known. Studying this dependency and quantifying it through LES simulation is a fascinating topic that could be a stand-alone paper.

• There exist only a few studies of the coherence of turbulence above the atmospheric surface layer (ASL) (e.g. Lothon et al., 2006). A large portion of the rotor plane of modern offshore wind turbines is located above the ASL. How to scale the coherence of turbulence above the ASL? Can the depth of the atmospheric boundary layer influence the coherence above the ASL? LES could give some insight into these questions. This topic is maybe more related to atmospheric science, but it could still be relevant to wind energy science.

**Reply**: Our goal with the abstract was to point out future applications, not necessarily say that we demonstrated that work in the current manuscript. We have updated the wording on the abstract to reflect that. We agree that the three points raised are relevant for future research. Our goal with the manuscript was to be general, focusing more on details of the first point (lateral coherence), while also acknowledging some of the other two points raised. We had showed the coherence at lateral and vertical separations, but had not established the link between the two; we have now added Fig. 12, explicitly comparing the two, with an accompanying discussion. That is relevant, but given that the work was based on a single atmospheric stability state, we refrained from claiming any general conclusion about the relationship, and opted to leave the results as obtained.

These points were all mentioned and were covered relatively superficially because our intent was to highlight that LES is indeed powerful and is able to capture the physics needed to improve simpler models. We wanted to show the potential in model improvement, which was also the reason why we mentioned structural loading as a potential discipline that could benefit from better understanding of the coherence characteristics. We have also added details about other areas that can benefit from accurate coherence estimates: lateral coherence in the v and w component are needed for dynamic wake meandering; longitudinal coherence are informative for control strategies that use feed-forward approaches. We have added some discussion around the points you raised, about the other areas where the tools displayed can be used, and tried to make the goal of the paper more clear.

We appreciate your high-level comments about the paper and we hope the revised manuscript makes a more clear point about the goal of the paper. Please find below our responses to the specific comments submitted as Community Comments.

**Specific comments from Reviewer 1 submitted as Community Comments**

**Reviewer Point P 1.4** — Abstract: I feel the abstract is somewhat self-contradictory. The authors write first that turbulence statistics and spectral analysis give limited information to describe a turbulent flow. Then the authors state that they focus on the coherence and spatial correlation functions. But the coherence is also a (two-point) spectral characteristic of turbulence. The integral time scale and integral length scale are also turbulence characteristics. So I do not really understand the logical reasoning here. (The turbulence length scales are integral characteristics and contain less information than the coherence or auto- and cross-covariance functions)

**Reply**: Thanks for pointing that out. The reviewer is right. We have reworded the first sentence to reflect what we meant to say: that typically one-point turbulence statistics, such as means and Reynolds stresses, are the quantities reported in studies that focused on microscale flow features. Correlation, coherence, and quantities derived from those are much less commonly computed.

**Reviewer Point P 1.5** — Line 23-25: It should be noted that the coherence does not necessarily compare the phase relationship between two time series or spatial series. In eq. 3, the authors use

the magnitude-squared coherence (mscoh) which does not give any information on the phase. This is something they judiciously comment. A more appropriate definition of coherence is given by Ropelewski et al (1973), which states that the coherence is a correlation function in the frequency space.

**Reply**: We have removed the reference to phase. As you mentioned, we do comment on the phase and the coherence definition (mscoh) we used in the paragraph following Eq 3. Thanks for catching that.

**Reviewer Point P 1.6** — Line 31-32: I understand what the authors mean with "implicitely", but this sentence is actually incorrect. In the Davenport model (Davenport, 1962), the separation distance is explicitly needed (and modelled) in the coherence function:

$$\gamma_u = \exp\left(\frac{-C_z^u d_z f}{\bar{u}}\right) \tag{1}$$

where  $d_z$  is the vertical separation distance, f is the frequency,  $C_z^u$  is an empirical decay coefficient and u is the mean wind speed. Interestingly, in the Davenport model, the dependency of  $C_z^u$  on the separation distance is not modelled. This dependency was found to be, sometimes, necessary for vertical separations (Bowen et al., 1983; Cheynet, 2019) and lateral spatial separations (Sacré and Delaunay, 1992; Cheynet et al., 2017).

**Reply**: We have corrected the mistake and expanded the paragraph, including the equation and citations you mentioned. Thanks for the suggestion.

**Reviewer Point P 1.7** — Line 33: Contrary to a popular belief, Kaimal et al. (1972) did not study the coherence of turbulence. Therefore, writing "Kaimal's coherence" is incorrect. A more appropriate wording for this coherence model would be "exponential decay model" based on the Davenport model. It can also be noted that the empirical velocity spectra model used in IEC 61400-1 (2005) is not Kaimal's model either, although the work by Kaimal et al. (1972) was used as a basis for this model.

**Reply**: We have completely rewritten the paragraph and added more information. We have also corrected references to "Kaimal's model" throughout the manuscript.

**Reviewer Point P 1.8** — Line 41: We should note that in the marine atmospheric boundary layer (MABL), non-neutral conditions are also commonly observed at wind speed above the rated wind speed of wind turbines (Barthelmie, 1999; Sathe et al., 2011; Cheynet et al., 2018).

**Reply**: Thanks for the references. We have removed the "for low wind speed scenarios" part of the sentence.

**Reviewer Point P 1.9** — Line 53-54: The sentence "In general, these studies found that coherence levels based on observations at an offshore environment are higher than those computed by the spectral model" may be contradicted by the scientific literature. In Cheynet et al. (2018), the IEC exponential coherence model is found to model fairly well to the vertical coherence under near-neutral conditions at FINO1. However, as for the Davenport model, the IEC exponential coherence model has two main drawbacks: (1) it does not account for the thermal stratification

of the atmosphere and (2) its main decay coefficient, which is equal to 12, is independent on the separation distance. In Mann (1994) and Cheynet (2019), the uniform-shear (US) model tends to slightly overestimate the vertical coherence of turbulence. It can be noted that, following Mann (1994), the US model captures remarkably well the lateral coherence of near-neutral turbulence.

**Reply**: We have missed your study from 2018 that found a good match between observations and IEC model. We have modified the quoted sentence to account for this study and have added a citation.

**Reviewer Point P 1.10** — Page 2: I agree with the authors that studying the lateral coherence with sonic anemometers is challenging. However, anemometers mounted on bridges in coastal areas (e.g. Sacré and Delaunay, 1992; Kristensen and Jensen, 1979) have been used in the past with encouraging results. For the past few years, scanning Doppler wind lidars have also been used to study the lateral coherence of turbulence (with relative success). Sonic anemometer data from bridges, while not offshore, are still valuable to validate spectral turbulence models. Also, the US model Mann (1994) was validated for lateral separation in the MABL. This may be worth mentioning in the manuscript.

Reply: We have added a discussion on this and cited a few works. Thanks for pointing it out.

**Reviewer Point P 1.11** — Line 162-163: A more robust way to test Taylor's assumption of frozen turbulence is to study the coherence of turbulence with longitudinal separations. If the turbulent field is frozen, the longitudinal coherence will be equal to 1 at every frequency. In reality, it will not be the case This method can also be used to show that smaller eddies will not satisfy the assumption of frozen turbulence at separation greater than a few dozen of meters while bigger eddies may be considered as "frozen" over a distance greater than e.g. 100 m (Cheynet et al., 2017, Fig. 11). The approach adopted in Fig 6 is more limited as it only focuses on large eddies.

**Reply**: Thanks for the comment. We understand the limitations of the approach used on Fig. 6. Here we were not aiming in showing that frozen turbulence was valid for all scales. In fact, we note that our the grid resolution is unable to properly resolve the smaller eddies. As the focus of the coherence analysis was on the low-frequency large-scale range of scales, we feel the comparison shown in Fig. 6 to be appropriate.

**Reviewer Point P 1.12** — The paper seems to dedicate a significant part to the coherence at longitudinal separation. From the viewpoint of wind turbine design, this may be a minor challenge compared to the need to reduce the uncertainties associated with the co-coherence at lateral and vertical separations.

**Reply**: We agree that from a wind turbine design point of view, the longitudinally separated coherence is not nearly as important as the laterally and vertically separated. Our goal in showing all three separation directions was to highlight that, even though such quantity may not be as important for turbine design, it is important for other disciplines (e.g., controls). Additionally, we wanted to point out that the use of LES allows you to obtain such a quantity just as easily as the other separation directions. We also took the opportunity to compare the longitudinally separated results with a prior model from Simley et al. (Fig. 9), developed with inflow preview-based control strategies in mind. Finally, in the authors' opinion, it is also instructive to look at Fig. 13 and see the longitudinal direction with significantly higher

levels of coherences, which is both intuitive and worthwhile to see compared side by side with the other components.

**Reviewer Point P 1.13** — Line 240: The relationship between wind loading and wind coherence was developed and modelled in the 1960s by pioneers such as Davenport (Davenport, 1962), through the so-called "buffeting theory". This theory was originally applied to high-rise buildings and bridges. Maybe a lesser-known fact is that the response of a wind turbine can be quantitatively described without an elaborated numerical model by the buffeting theory. I recommend reformulating line 162 by stating that the theory related to the wind-induced response of wind turbines can be traced back to the 1960s with the development of the buffeting theory, where both the coherence and one-point velocity spectra were used to predict the dynamic response of wind-sensitive slender structures.

**Reply**: It is unclear what specific line the reviewer is referring to, as we believe the reviewer does not mean line 162. Thanks for the suggestion- we have included the note about Davenport's buffeting theory.

**Reviewer Point P 1.14** — Line 263-265 and Lines 270-271: It should be kept in mind that one fundamental assumption to study and model the coherence of turbulence is that the flow is fairly homogeneous and stationary. Downstream of a wind turbine and inside a wind farm, the flow can be strongly heterogeneous. Therefore, the paragraph on lines 263-265 and lines 270-271 may be criticized for violating the assumption of flow homogeneity. I think these lines can be removed without problem.

**Reply**: We have removed those lines and added a discussion on the logic behind computing the longitudinal coherence. See the response of the next point.

**Reviewer Point P 1.15** — Line 268-270: I do not understand the logical reasoning behind the sentence "The coherence values for laterally separated points [...] suggest that it may be important to properly account for the longitudinal separation, in addition to the lateral and vertical separations included in the IEC standard". The fact that the lateral coherence  $coh_y$  is smaller than the longitudinal coherence cohx does not justify the need to model cohx for wind loading on a wind turbine. If turbulence is considered frozen, which is commonly done, then for a given separation,  $coh_x \ge coh_y$  at every frequency.

**Reply**: We appreciate you pointing this out. We agree that  $coh_x$  is not important for loads. We were writing this part also with control strategies in mind. For instance, a few studies have looked at longitudinal coherence upstream of the rotor for controls purposes (e.g. [2, 6, 7, 4]). We have reworded the paragraphs surrounding the mentioned lines. We also removed the reference to the standard and added a discussion on the longitudinal coherence for controls, which ties back to the previous point raised.

**Reviewer Point P 1.16** — Line 272-273: The fact that the coherence is not always equal to 1 at zero frequency was documented by Kristensen and Jensen (1979) and in many studies since then. This should be mentioned in the manuscript

Reply: Thanks for the suggestion. We have added a sentence noting that has been a known fact.

**Reviewer Point P 1.17** — Line 274-276: The fact that the mscoh does not converge toward zero is not due to numerical noise but because the mscoh is a biased estimator. This bias is documented in e.g. Kristensen and Kirkegaard (1986) and commented in Mann (1994, p. 156). To avoid (or limit) this bias, I recommend focusing on the co-coherence.

**Reply**: We appreciate the references given. The work cited used an analogous method as we did in this work and attributed the lack of convergence towards zero to numerical noise. We have added a note about the bias and cited the reference you gave, but also left the reference of Shaler et al. We modified the sentence making it explicit what was given in each of the two works cited.

**Reviewer Point P 1.18** — It should be noted that experimental studies of the coherence at separation distance greater than 100 m are associated with significant uncertainties because only a few data points are different from zero or are hidden in the ambient noise. To reduce such uncertainties, a large number of samples should be used. The coherence estimates will be then ensemble averaged. Ideally, both short and large separation distances should also be considered to capture the full extent of the coherence. If such conditions are not met, it can be challenging to reach meaningful conclusions.

**Reply**: We agree with your comment. Unfortunately, we only had data from a handful of instruments, so in order to create more realizations for an ensemble average, we used a windowing approach. Choosing an appropriate window on this varying-condition dataset was a balance between more realizations and appropriate window size (as pointed out in [3] as the choice of number of windows being "a matter of compromise").

**Reviewer Point P 1.19** — Page 14: I do not recommend using negative separations because these have no physical meaning and could trigger numerical errors with exponential coherence models. Instead, I suggest focusing on pairs of measurement heights or lateral positions as formulated in Putri et al. (2022, p. 1697).

**Reply**: We agree that negative separation could trigger numerical errors if just plugged into the exponential-based models. We also understand the lack of physical meaning. When we talk about negative separation, we always follow with a comment talking about the negative or positive indicating directionality up or down, including the caption of Fig. 11–13. It is typical to talk about "separation" distances in the context of coherence, but we wanted to emphasize that (in the case of this paper), points located at heights of 40 and 80 m are different than points at 80 and 120 m, even though their separation distances are clearly the same. The discussion of negative separation appears on the vertical coherence, which comes after the discussions about longitudinal and lateral coherence, where the term "separation" had been used. We feel like switching to another definition of the two points at this point could introduce unnecessary complexity and make it harder for a clear, direct comparison of vertical and lateral coherences. We have added some notes about the negative separation and made sure to note what is means at every mention of it.

**Reviewer Point P 1.20** — Line 280-288: From my understanding, the asymmetry effect mentioned by the authors has been studied in detail in the MABL by Cheynet (2019) at FINO1 and by Putri et al. (2022) near Vindeby wind farm. The asymmetry can be modelled quite well using the approach adopted by Bowen et al (1983) and reflects the presence of the ground, which blocks the flow. Some interesting questions are (1) how this asymmetry changes with the thermal stratification of the atmosphere and (2) how it affects the loading on a wind turbine. These challenges can be discussed by the authors or be the topic of further study.

**Reply**: This is a good point; thanks for the references. We have added more discussions about the asymmetry aspect in the paragraph you mentioned and also in the discussion section.

**Reviewer Point P 1.21** — Line 288-289: The justification with reference to Naito (1983) seems incorrect to me. The coherence is lower than unity at zero frequency because the separation distance is not negligible compared to a typical length scale of turbulence (Kristensen and Jensen, 1979). If there exist long period fluctuations, then the time series may be non-stationary and prevent the study of turbulence characteristics. Fortunately, a quick look at Fig 2 suggests that the time series are reasonably stationary.

**Reply**: The work of Naito attributed the fact that the coherence does not approach unity for v and w to be "caused by [these] components scarcely include fluctuations of long periods in the surface layer". We left that sentence in, being more clear that the author attributed to such reason, but also included your suggestion based of the work of Kristensen and Jensen.

**Reviewer Point P 1.22** — Line 332-334: The suggestion "Future work should include a parametric study with varying atmospheric conditions, which will produce a set of coherence curves for each component and separation direction" has been done for vertical separations in Cheynet et al. (2018) at FINO1 using two years of continuous measurements. More generally, the dependency of the coherence on the atmospheric stability is also documented for an onshore location in Soucy et al. (1982) and in the MABL by Putri et al. (2022).

**Reply**: The reviewer is right, this has been done in several other studies. We have rephrased the discussion surrounding the mentioned lines to highlight the limitations of the other studies (streamwise component; limited amount of separation distances). We have added a couple of citations too.

**Reviewer Point P 1.23** — Line 339-344: I agree with the authors. However, this paragraph is partly covered by the buffeting theory that was developed in 1960s and 1970s. So I am not sure if it is still an up-to-date material for discussion. Maybe this can be elaborated or removed?

**Reply**: The authors are not totally sure what the reviewer meant. Is the buffeting theory mentioned in the sense of steady flow producing dynamic loading, perhaps related to deep stall regimes, or related to the slender structures from Davenport's work [1]? We touched on that in our response to point P 1.13. In any case, no claims are made about when those findings were first published.

**Reviewer Point P 1.24** — Section 5: One of the limiting aspects of the coherence for stochastic flow simulation is the need for a fairly homogeneous flow, which is not the case inside a wind farm. This limit can be partly overcome by LES simulation. maybe this aspect can be discussed in more detail here?

**Reply**: We agree that this limit can be partially overcome by LES. However, if performing an LES of the entire wind farm, coherence characteristics are automatically accounted for because the turbines are explicitly modeled within the flow field. One typical workflow based on stochastic inflow generation

with coherence modeling involves first generating a background flow, then applying it in a separate tool (e.g., FAST.Farm, as done in Shaler et al.[5, 8]). We have added a quick discussion on further uses of stochastic inflow generated with LES-informed exponential coherence models to Section 5. We want to emphasize that this work provides guidance for follow-on work, allowing the study of coherence characteristics from point to point in the flow, in ways that are not feasible with instruments.

**Reviewer 2**

The authors thank reviewer 2 for his/her comments. We have incorporated all suggestions into the manuscript. Find below our responses to specific points raised.

**Reviewer Point P 2.1** — Line 27: is the description of "exponential-based equations" really applicable for Mann's model?

Reply: The reviewer is right. Thanks for catching that. We have re-phrased the surrounding sentences.

**Reviewer Point P 2.2** — Line 43: The IEC standards give the 90th percentile TI, which will naturally be quite conservative when compared to more commonly observed conditions.

Reply: Right. We have added a sentence to make that more explicit.

**Reviewer Point P 2.3** — You might also find some additional studies of the effects of coherence on dynamic wake meandering for floating wind turbines interesting (example: Wise and Bachynski 2020).

**Reply**: Thanks for this comment and suggestion. We have added a discussion on the studies involving coherence within the context of wake meandering. That was an interesting discussion that we did not have in the manuscript that makes a strong case for v and w coherences. We included the reference given and added discussion to multiple sections.

**Reviewer Point P 2.4** — Line 61-62: Although the Mann and Kaimal models don't capture the effect of atmospheric stability on shear and turbulence levels, do they capture the effects of shear and turbulence on the coherent structures?

**Reply**: We have rephrased a few sentence surrounding the lines mentioned to avoid ambiguous statements and make the text more clear.

**Reviewer Point P 2.5** — The difference in the mean wind speed is almost 2 m/s at 80 m, which is described as slight. How large of a difference would you consider to be large?

**Reply**: Thanks for noting that. It is indeed difficult to say what is "slight" and what is "large". For power curve considerations, 2 m/s is large, but for error in mesoscale models compared to real data, 2 m/s may be small. We have removed the word "slight" from the sentence.

**Reviewer Point P 2.6** — Line 126: little realizations – do you mean "few realizations"?

**Reply**: Yes. Thanks for catching that. Fixed.

**Reviewer Point P 2.7** — Are the streamwise and crosswise directions defined for each 15 min window, or instantaneously? Could the results in Fig. 3 and 4 alternatively be shown such that x corresponds to the mean wind direction in each 15 min window? I'm not sure that I completely understand whether the black and magenta arrows are indicating an average over the full time period or if the fields have been aligned to these directions in each time interval.

**Reply**: The questions raised are fair and we acknowledge more details about the image and the window procedure should have been given. We have updated the images and descriptions to make it more explicit what exactly was done.

For every 15-min window, we obtain a mean wind direction. That wind direction is used to obtain the correlation coefficients from the maps shown in Figs. 3 and 4. The curves are then presented in Fig. 5. In Fig. 5, each light shade curve represents one 15-min interval, and each curve is the mean of 15-min worth of snapshots (at 1 Hz), *sampled along the same (interval-mean) wind direction*. To facilitate the understanding of the procedure, we have removed the arrows from Fig. 3, and added to the text the general wind direction, referencing the reader back to the background conditions shown in Fig. 2. For Fig. 4, instead of showing the ensemble average in space and time with the averaged wind direction, we now opted to show a single 15-min interval, where the relationship between the arrow and the actual data is more clear.

What the reviewer is asking regarding the x direction corresponding to the mean wind direction is exactly what is shown in Fig. 5, in a more quantitative way. Because of that, we preferred to leave Figs. 3 and 4 without any rotation.

**Reviewer Point P 2.8** — Line 192: In order to see the effect of mesoscale transients, it would be useful to compare to a case without transients.

**Reply**: That is a good point. A case without transients would no longer contain mesoscale information. In this study, while the conditions are changing, the selected day and period is relatively stationary, with no sudden change in conditions (like what would be experienced in a coastal front, for instance). Our aim was to demonstrate a general mesoscale-coupled approach that could be applied under both stationary and non-stationary conditions. Also, note that when we refer to transient, we do not mean we are looking at the actual transients in the sense of a response to a certain (step) change. We are interested in looking at how the coherence changes over different periods with different conditions, and not exactly how they perform during the transient.

**Reviewer Point P 2.9** — Line 247: It would be good to specify the lowest frequency of interest (not just the highest). I would also suggest replacing "about" with "approximately" in this sentence.

**Reply**: Thanks for the suggestions. "About" has been replace. We have also added a sentence talking about the lowest frequencies from the power spectral density plots and relating that to the axis limits shown in Fig. 9.

**Reviewer Point P 2.10** — It would be good to be careful with the descriptions of the models, as "Kaimal" usually refers to the spectrum, while the coherence model is exponential.

**Reply**: Thanks for pointing this out. We have revised the entire manuscript making sure the descriptions are appropriate. Small changes throughout the manuscript can be seen in the marked-up pdf file.

**Reviewer Point P 2.11** — Line 278: 80 m seems low as a typical hub height offshore.

**Reply**: Yes, we agree. We have revised the manuscript removing the reference to a hub height and instead mention a "measurement point at 80 m". We added a sentence explaining why 80 m. The reason is that we had data available at (approximately) 40, 60, and 80 m from the FINO1 platform. All of our analysis was conducted at 80 m because of data availability.

**Reviewer Point P 2.12** — Line 310: awkward phrasing as it becomes unclear what is reported in the literature.

**Reply**: Thanks for catching this. We agree the sentence could be better. We have revised it being more explicit.

**Reviewer Point P 2.13** — Line 331: Isn't the coherence in TIMESR based on the Davenport model?

**Reply**: TurbSim allows the creation of the synthetic flowfield to be based on a time-series of a single point, which is the option the reviewer is referring to, TIMESR. Another option in Turbsim, which is what the authors are referring to, is the IECKAI as the TurbModel, which applies the Kaimal exponential model based on the coherence values given in the variables related to the "spatial coherence parameters" within the input file. The default values are what based on the standard (which varies by version of the standard, but the desired version can be given in yet another variable in the TurbSim input file).

**Reviewer Point P 2.14** — Line 365: I think it would be more accurate to say that the IEC standard does not include information about coherence in the v and w directions, rather than the model ignores them. Exponential coherence in these directions can also be applied, as shown by i.e. Shaler et al.

**Reply**: Thanks for pointing this out. We have rephrased the sentence and added more details about this point.

**Reviewer Point P 2.15** — Similarly, in the discussion, it would be good to distinguish between how models have been applied and how they can be applied – and also keep in mind that design standards are generally intended to introduce some conservatism.

**Reply**: We appreciate the suggestion. We have expanded the discussion including your comments. One of the goals of this paper was indeed to note that design standards are sometimes *too* conservative and we could use high-fidelity modeling to improve that. We thank the reviewer once again for all of the useful comments.

**References**

- [1] Alan G Davenport. The response of slender, line-like structures to a gusty wind. *Proceedings* of the Institution of Civil Engineers, 23(3):389–408, 1962.
- [2] Mithu Debnath, P Brugger, Eric Simley, Paula Doubrawa, Nicholas Hamilton, Andrew Scholbrock, David Jager, Mark Murphy, Jason Roadman, Julie K Lundquist, et al. Longitudinal coherence and short-term wind speed prediction based on a nacelle-mounted doppler lidar. In *Journal of Physics: Conference Series*, volume 1618, page 032051. IOP Publishing, 2020.
- [3] Leif Kristensen and Peter Kirkegaard. Sampling problems with spectral coherence, volume 526. Risø National Laboratory Roskilde, Denmark, 1986.
- [4] David Schlipf, Po Wen Cheng, and Jakob Mann. Model of the correlation between lidar systems and wind turbines for lidar-assisted control. *Journal of Atmospheric and Oceanic Technology*, 30(10):2233–2240, 2013.
- [5] Kelsey Shaler, Jason Jonkman, Paula Doubrawa Moreira, and Nicholas Hamilton. FAST.Farm response to varying wind inflow techniques. Technical report, National Renewable Energy Lab.(NREL), Golden, CO (United States), 2019.
- [6] Eric Simley, Nikolas Angelou, Torben Mikkelsen, Mikael Sjöholm, Jakob Mann, and Lucy Y Pao. Characterization of wind velocities in the upstream induction zone of a wind turbine using scanning continuous-wave lidars. *Journal of Renewable and Sustainable Energy*, 8(1):013301, 2016.
- [7] Eric Simley and Lucy Y Pao. A longitudinal spatial coherence model for wind evolution based on large-eddy simulation. In 2015 American Control Conference (ACC), pages 3708–3714. IEEE, 2015.
- [8] Adam S Wise and Erin E Bachynski. Wake meandering effects on floating wind turbines. Wind Energy, 23(5):1266–1285, 2020.

---

## Referee Report (RR1)

**Investigations of Correlation and Coherence in Turbulence from a Large-Eddy Simulation - Version 2**

Reviewer's comments

**General comment**

The revision of the manuscript "Investigations of Correlation and Coherence in Turbulence from a Large-Eddy Simulation" by Thedin et al. has substantially improved since the previous version. The nice explanation of the authors in their replies to the reviewers has also helped me a lot to understand the objective of the paper. Most of my comments are specific and, fortunately, can be addressed through **minor revisions**.

For the general comment, I recommend the authors split section 4 "Methodology & results" into two independent sections: the first one would be named "Methods" and the second one would be named "Results". In the current version, section 4 is too unstructured to allow the reader to easily grasp the logical pattern of the results. The new section "Methods" should contain the necessary information on (1) the numerical setup, (2) the method to derive the spatial, temporal correlation and spectral characteristics and (3) the background information on the previous turbulence models.

**Specific comments**

**Point 1**

Line 1 and line 21: The contradiction that was noted in the previous version (Point 1.4, version 1) is still present in the abstract and one line 21. To remove the contradiction, I suggest simply removing "and/or spectral analysis" in line 2 and line 21. The reason is that coherence analysis is part of spectral analysis. Also, auto-correlation analysis is directly connected to a spectral analysis by the Wiener–Khinchin theorem. The latter states that for a stationary random process, the power spectrum of the process is the Fourier transform of the autocorrelation function. More generally, lines 1 and 2 could be formulated as "Microscale flow descriptions are often given in terms of integral flow characteristics. Those metrics, while valuable, give limited information about the spatial and temporal structure of turbulent eddies."

**Point 2**

Line 34: The equation for the Davenport coherence model can be given in a new line with an equation number. This is a little more elegant than an in-line equation.

**Point 3**

Line 49: We should remember that "Kaimal's exponential decay model" is inaccurate since the exponential decay model is actually from Davenport. In the technical report by Thresher et al. (1981) the authors use the expression "Davenport-Kaimal model", which is much fairer since it indicates the combination of the Kaimal spectral model with the Davenport coherence model. Alternatively, the term "Thresher's model" could be used too.

**Point 4**

Line 72: The reference to Wise and Bachynski (2019,2020) and Shaler et al. (2019) should be written as a parenthetical citation rather than an in-text citation.

**Point 5**

Line 76: I do not understand the sentence "As mentioned, standard only specifies in the streamwise direction". Do you mean "As mentioned, the IEC standard only specifies the coherence of the along-wind component in the cross-wind direction"?

**Point 6**

Line 70-77: I was unaware of this information. This is quite useful! I am a little puzzled by the choice of some authors to have a fully correlated wind field for the v and w velocity components if the standard does not explicitly state which coherence values should be used. In wind engineering, the coherence of the three velocity components (u, v and w) is usually modelled, even though there exist a lot of uncertainties. The review by Solari and Piccardo (2001) is quite enriching in this regard.

**Point 7**

Line 87: For the sake of clarity, I suggest reformulating the last sentence as "In the IEC standard, the thermal stratification of the atmosphere is not explicitly accounted for by either the Mann or Davenport-Kaimal model". In practice, it is possible to (partly) account for the stability in the Mann model by fitting this model to in-situ data representative of unstable or stable conditions as done by Sathe et al. (2013). The same idea applies to the Davenport model, see e.g. Soucy et al. (1982); Cheynet et al. (2018), where the Davenport decay coefficient become stability-dependent for the three velocity components.

**Point 8**

Lines 88-93: This paragraph is really nice and, I believe, crucial to the understanding of the paper. I suggest moving it to the beginning of the introduction, typically after the first or second

paragraph. In general, the objectives of the study should be announced early. Also, a new paragraph announcing the structure of the paper can be added at the end of the section "Introduction".

**Point 9**

Lines 105: I suggest removing "and will only be as accurate as the mesoscale". This part is contradicted by the sentence coming immediately after, since the microscale gives information on turbulence but not the mesoscale.

**Point 10**

Figure 2: This is a nice and clear figure. Maybe one sentence can be added to explain how the wind shear exponent is calculated. This would be useful to the reader. This sentence could be placed in the section "Methods".

**Point 11**

Section 4.3: For the sake of pedagogy, it could be briefly mentioned that since the flow is assumed homogeneous, spatial averaging is equivalent to ensemble averaging. Or maybe has it been already mentioned?

**Point 12**

Section 4.3 bis: I really like the idea to assess Taylor's hypothesis of frozen turbulence by using the integral time scale and integral length scale. For the sake of clarity, I suggest writing the equations demonstrating how these quantities are calculated. In particular, the estimation of the integral length scale (or time scale) can be obtained either by (1) integration of the auto-correlation down to the first zero crossing or (2) by modelling the autocorrelation with an exponential decay as

$$R_u(d_x) = \exp\left(\frac{-d_x}{L_u^x}\right) \tag{1}$$

where  $R_u(d_x)$  is the auto-correlation function;  $d_x$  is the streamwise separation distance and  $L_u^x$  is the integral length scale of the *u*-component in the *x*-direction.

**Point 13**

Figure 8: For the sake of clarity, replacing "integral length scale" with a symbol may be preferable. For example, the previous point uses  $L_u^x$  which specified both the direction and velocity component. The integral length scales of the along-wind component could be  $L_u^x$  (streamwise separation)  $L_u^y$  (lateral separation) or  $L_u^z$  (vertical separations).

**Point 14**

Line 248: It is true that Nybø et al. use the term "co-coherence" and "quad-coherence". However, their paper is not a primary source. The possible primary source is the thesis by Watson (1975). The thesis is openly available at this link. The term "co-coherence" was further used in the 1980s by Barnard (1981) among others.

**Point 15**

Line 251: I think "Kaimal exponential coherence model" can be replaced by "IEC exponential decay model".

**Point 16**

Line 260-261: "representing second-order statistics" may be removed since the Mann model also describes second-order statistics only.

**Point 17**

Line 275: I suggest reformulating "Nowadays, more complex simulation tools" into "Nowadays, simulation tools for wind energy application". Indeed, engineering tools relying on the Davenport model can be considerably more complex than the IEC standard, for example, the ESDU standards for the Wind Engineering Series.

**Point 18**

Line 367: This line read as "Fig 14 summarizes the importance of modelling all three components of the turbulence". Is it the importance for wind loading or wake meandering?

**Point 19**

Fig 14: Is this figure necessary to the paper? If the authors decide to keep it, I suggest not using a contour map but a pseudocolour plot instead. The contour lines introduce artefacts that can lead to misinterpretations.

**Point 20**

Line 422: The discussion is interesting, but I suggest not discussing the cut-off frequency in terms of frequencies (Hz) but in terms of wavenumbers  $(m^{-1})$ . Otherwise, the cut-off frequency will depend on the mean wind speed.

**Point 21**

Line 448-449: the part "suggesting that frozen turbulence may not be applicable under other conditions" may be reformulated more clearly. If the hypothesis of frozen turbulence is discussed in terms of coherence, it should be related to the size of eddies. For example, at a specific spatial separation, the turbulence can be considered frozen for large eddies (high coherence) but not for small eddies (low coherence).

**Point 22**

Line 452: "better inform turbulence models" is a little unclear to me. Maybe "improve turbulence models" is better a better formulation?

**References**

- Thresher, R., Holley, W., Smith, C., Jafarey, N., Lin, S.. Modeling the response of wind turbines to atmospheric turbulence. Tech. Rep.; Oregon State Univ., Corvallis (USA). Dept. of Mechanical Engineering; 1981.
- Solari, G., Piccardo, G.. Probabilistic 3-D turbulence modeling for gust buffeting of structures. Probabilistic Engineering Mechanics 2001;16(1):73–86.
- Sathe, A., Mann, J., Barlas, T., Bierbooms, W., Van Bussel, G.. Influence of atmospheric stability on wind turbine loads. Wind Energy 2013;16(7):1013–1032.
- Soucy, R., Woodward, R., Panofsky, H.. Vertical cross-spectra of horizontal velocity components at the Boulder observatory. Boundary-Layer Meteorology 1982;24(1):57–66.
- Cheynet, E., Jakobsen, J., Reuder, J.. Velocity spectra and coherence estimates in the marine atmospheric boundary layer. Boundary-Layer Meteorology 2018;169(3):429–460.
- Watson, B.H.H.. A study of the statistical approach to wind loading. Master's thesis; University of Cape Town; 1975.
- Barnard, R.. Wind loads on cantilevered roof structures. Journal of Wind Engineering and Industrial Aerodynamics 1981;8(1-2):21–30.

---

## Author Response (AR2)

**WES 2022–71: Author's Response to the Reviewers (Version 2)**

The authors thank both reviewers for their further comments. A marked-up version of the manuscript highlighting all the changes is available. Note that large chunks highlighted in the marked-up version are related to re-organization of the manuscript.
* * *
**Reviewer 1, Etienne Cheynet**

The authors thank Dr. Cheynet for his time in reviewing our revised manuscript. We appreciate the further comments provided and have addressed them below.

**Reviewer Point P 1.1** — The revision of the manuscript "Investigations of Correlation and Coherence in Turbulence from a Large-Eddy Simulation" by Thedin et al. has substantially improved since the previous version. The nice explanation of the authors in their replies to the reviewers has also helped me a lot to understand the objective of the paper. Most of my comments are specific and, fortunately, can be addressed through minor revisions.

For the general comment, I recommend the authors split section 4 "Methodology & results" into two independent sections: the first one would be named "Methods" and the second one would be named "Results". In the current version, section 4 is too unstructured to allow the reader to easily grasp the logical pattern of the results. The new section "Methods" should contain the necessary information on (1) the numerical setup, (2) the method to derive the spatial, temporal correlation and spectral characteristics and (3) the background information on the previous turbulence models.

**Reply**: Thank you for the comments. We agree that the organization of the paper could be improved by separating methods and results. We have moved the sections around and re-structured it into two separate sections, also including the "Scenarios investigated" section into the new "Methodology".

**Reviewer Point P 1.2** — Line 1 and line 21: The contradiction that was noted in the previous version (Point 1.4, version 1) is still present in the abstract and one line 21. To remove the contradiction, I suggest simply removing "and/or spectral analysis" in line 2 and line 21. The reason is that coherence analysis is part of spectral analysis. Also, auto-correlation analysis is directly connected to a spectral analysis by the Wiener–Khinchin theorem. The latter states that for a stationary random process, the power spectrum of the process is the Fourier transform of the autocorrelation function. More generally, lines 1 and 2 could be formulated as "Microscale flow descriptions are often given in terms of integral flow characteristics. Those metrics, while valuable, give limited information about the spatial and temporal structure of turbulent eddies."

**Reply**: Thank you for the suggestion. We meant to say that simple spectral analysis is usually done just to ensure the cascade follows $-5/3$. In any case, your point is valid and we have removed that mention from the abstract and the text.

**Reviewer Point P 1.3** — Point 2 Line 34: The equation for the Davenport coherence model can be given in a new line with an equation number. This is a little more elegant than an in-line equation.

**Reply**: Done.

**Reviewer Point P 1.4** — Line 49: We should remember that "Kaimal's exponential decay model" is inaccurate since the exponential decay model is actually from Davenport. In the technical report by Thresher et al (1981) the authors use the expression "Davenport-Kaimal model", which is much fairer since it indicates the combination of the Kaimal spectral model with the Davenport coherence model Alternatively, the term "Thresher's model" could be used too.

**Reply**: We have made small updates throughout the document to address this issue. Thanks for highlighting it once again.

**Reviewer Point P 1.5** — Line 72: The reference to Wise and Bachynski (2019,2020) and Shaler et al. (2019) should be written as a parenthetical citation rather than an in-text citation.

**Reply**: Yes. That was a mistake on our part. Fixed it.

**Reviewer Point P 1.6** — Line 76: I do not understand the sentence "As mentioned, standard only specifies in the streamwise direction". Do you mean "As mentioned, the IEC standard only specifies the coherence of the along-wind component in the cross-wind direction"?

**Reply**: We do mean that the IEC standard only specifies the coherence of the streamwise component, in the vertical and crosswind direction. We made the sentence more clear. We revised the entire manuscript so that streamwise and cross-stream refer to the components and along-wind and crosswind refer to directionality.

**Reviewer Point P 1.7** — Line 70-77: I was unaware of this information. This is quite useful! I am a little puzzled by the choice of some authors to have a fully correlated wind field for the v and w velocity components if the standard does not explicitly state which coherence values should be used. In wind engineering, the coherence of the three velocity components (u, v and w) is usually modelled, even though there exist a lot of uncertainties. The review by Solari and Piccardo (2001) is quite enriching in this regard.

**Reply**: We agree with your comment about some author's choice of coherence model in v and w. Some authors decide to use *a* equation to model the coherence in v and w, but no further work is done to assess the performance and accuracy. Unfortunately the latest version of the standard is quite confusing with the terms $L_c$ and $L_k$ which has resulted in inaccurate interpretations and thus inaccurate use of the suggestions from the standard.

**Reviewer Point P 1.8** — Line 87: For the sake of clarity, I suggest reformulating the last sentence as "In the IEC standard, the thermal stratification of the atmosphere is not explicitly accounted for by either the Mann or Davenport-Kaimal model". In practice, it is possible to (partly) account for the stability in the Mann model by fitting this model to in-situ data representative of unstable or stable conditions as done by Sathe et al. (2013). The same idea applies to the Davenport model, see e.g. Soucy et al. (1982); Cheynet et al. (2018), where the Davenport decay coefficient become stability-dependant for the three velocity components.

**Reply**: We have rephrased the sentence being more explicit about the "stability" aspect. We understand that such models can have their constants tuned/curve-fit using data representative of whatever stability state one wishes to represent. Our comment was more related to their original formulations– we made that more clear.

**Reviewer Point P 1.9** — Lines 88-93: This paragraph is really nice and, I believe, crucial to the understanding of the paper. I suggest moving it to the beginning of the introduction, typically after the first or second paragraph. In general, the objectives of the study should be announced early. Also, a new paragraph announcing the structure of the paper can be added at the end of the section "Introduction".

**Reply**: We are glad this paragraph is now clear and useful to the reader. We understand the value of having the objectives stated early on, but we have decided to keep it as is, since it functions a closing comment for the introduction, tying all the points discussed and clearly stating the objectives and what will come next.

**Reviewer Point P 1.10** — Lines 105: I suggest removing "and will only be as accurate as the mesoscale". This part is contradicted by the sentence coming immediately after, since the microscale gives information on turbulence but not the mesoscale.

**Reply**: When we say the microscale will only be as accurate as the mesoscale, we are referring to the mean quantities. Yes, the microscale will develop the lower scale turbulence, but it will not change the mean in any significant way. The point of the sentence is to say that the microscale may not match the observations, but will rather match the mesoscale—that is illustrated in Fig 2. We have added "mean quantities" to the sentence to make that more clear.

**Reviewer Point P 1.11** — Figure 2: This is a nice and clear figure. Maybe one sentence can be added to explain how the wind shear exponent is calculated. This would be useful to the reader. This sentence could be placed in the section "Methods".

**Reply**: Thanks for the suggestion. We have added some comments about it when discussing the figure.

**Reviewer Point P 1.12** — Section 4.3: For the sake of pedagogy, it could be briefly mentioned that since the flow is assumed homogeneous, spatial averaging is equivalent to ensemble averaging. Or maybe has it been already mentioned?

**Reply**: We have noted that on the spatial correlation subsection, in the context of using time averages as an ensemble average. But we agree that is not a bad idea to mention that again when referring to the spatial average. We added a comment about it in the beginning of the new "Results" section.

**Reviewer Point P 1.13** — Section 4.3 bis: I really like the idea to assess Taylor's hypothesis of frozen turbulence by using the integral time scale and integral length scale. For the sake of clarity, I suggest writing the equations demonstrating how these quantities are calculated. In particular, the estimation of the integral length scale (or time scale) can be obtained either by (1) integration of the auto-correlation down to the first zero crossing or (2) by modelling the autocorrelation with an exponential decay as

$$R_u(d_x) = \exp\left(\frac{-d_x}{L_u^x}\right) \tag{1}$$

where $R_u(d_x)$ is the auto-correlation function; $d_x$ is the streamwise separation distance and $L_u^x$ is the integral length scale of the $u$-component in the $x$-direction.

**Reply**: We computed the integral scales by numerical integration of the correlation curve we obtained (Fig. 7 in the latest version of the manuscript). We had mentioned that on line 219-221 of the first revision of the manuscript.

**Reviewer Point P 1.14** — Figure 8: For the sake of clarity, replacing "integral length scale" with a symbol may be preferable. For example, the previous point uses $L_u^x$ which specified both the direction and velocity component. The integral length scales of the along-wind component could be $L_u^x$ (streamwise separation) $L_u^y$ (lateral separation) or $L_u^z$ (vertical separations).

**Reply**: That is a fair point. We have tried to make it clear which direction and velocity components we used. We do not give any equations, but we have added symbols to make it easier to understand what quantity we are referring to in the larger context of integral scales and across different papers. We appreciate the suggestion.

**Reviewer Point P 1.15** — Line 248: It is true that Nybø et al. use the term "co-coherence" and "quad-coherence". However, their paper is not a primary source. The possible primary source is the thesis by Watson (1975). The thesis is openly available at this link. The term "co-coherence" was further used in the 1980s by Barnard (1981) among others.

**Reply**: Thanks for pointing that out. We did not mean to imply that Nibø et al coined the term, it was simply given as reference to the reader. We have removed the reference to Nibø et al's work.

**Reviewer Point P 1.16** — Line 251: I think "Kaimal exponential coherence model" can be replaced by "IEC exponential decay model".

**Reply**: As per the point P 1.4, we have modified these mentions throughout the manuscript to better reflect the authors of the models.

**Reviewer Point P 1.17** — Line 260-261: "representing second-order statistics" may be removed since the Mann model also describes second-order statistics only.

**Reply**: Done. Thanks for catching this.

**Reviewer Point P 1.18** — Line 275: I suggest reformulating "Nowadays, more complex simulation tools" into "Nowadays, simulation tools for wind energy application". Indeed, engineering tools relying on the Davenport model can be considerably more complex than the IEC standard, for example, the ESDU standards for the Wind Engineering Series.

**Reply**: We added the suggested "wind energy applications", but left "complex tools". Our reasoning is that some reasonably complex tools do make use of the models suggested by IEC (e.g. FAST.Farm).

**Reviewer Point P 1.19** — Line 367: This line read as "Fig 14 summarizes the importance of modelling all three components of the turbulence". Is it the importance for wind loading or wake meandering?

**Reply**: See response to next point.

**Reviewer Point P 1.20** — Fig 14: Is this figure necessary to the paper? If the authors decide to keep it, I suggest not using a contour map but a pseudocolour plot instead. The contour lines introduce artefacts that can lead to misinterpretations.

**Reply**: We were vague on purpose about this figure. It is important because it shows the coherence in the longitudinal separation which isn't usually given, as well as the v and w components being non-negligible. We decided to include it because it does give more "curves" than Figs. 9–11. We kept a contour map but have removed the thin lines between the levels– thanks for the suggestion.

**Reviewer Point P 1.21** — Line 422: The discussion is interesting, but I suggest not discussing the cut-off frequency in terms of frequencies (Hz) but in terms of wavenumbers ($m^{-1}$) Otherwise, the cut-off frequency will depend on the mean wind speed.

**Reply**: The cut-off frequency mentioned in the discussions was left in terms of frequencies because we wanted to refer to the specific setup we had and especially the plots shown. We understand that it depends on the mean wind speed, but it also depends on the grid resolution used. We believe the discussions are more consistent with the plots shown earlier if we talk in terms of the same quantities as before, as opposed to talk about in terms of wavenumbers for the first time in the paper.

**Reviewer Point P 1.22** — Line 448-449: the part "suggesting that frozen turbulence may not be applicable under other conditions" may be reformulated more clearly. If the hypothesis of frozen turbulence is discussed in terms of coherence, it should be related to the size of eddies. For example, at a specific spatial separation, the turbulence can be considered frozen for large eddies (high coherence) but not for small eddies (low coherence).

**Reply**: That's an interesting point. We added a few sentences relating it to the size of eddies. Thanks for the suggestion.

**Reviewer Point P 1.23** — Line 452: "better inform turbulence models" is a little unclear to me. Maybe "improve turbulence models" is better a better formulation?

**Reply**: We have changed the sentence to the suggested wording.
* * *
**Reviewer 2**

Thanks again to reviewer 2 for his/her comments. We have addressed all comments in the updated manuscript.

**Reviewer Point P 2.1** — Line 30: check subject-verb agreement

**Reply**: Fixed. Thanks for catching it.

**Reviewer Point P 2.2** — Line 61: the structures studied by Li et al. were lattice frames - not necessarily for wind turbines

**Reply**: We appreciate you catching that mistake. We have removed the incorrect reference.

**Reviewer Point P 2.3** — Line 265: "a sufficiently turbulent condition" - maybe "a sufficient description of the turbulence for load estimation purposes"?

**Reply**: We modified the sentence. Thanks for the suggestion.

**Reviewer Point P 2.4** — Line 282: lowest rather than highest? On a log scale, it's hard to say which frequencies are really correctly captured. For a 15 minute time series, the Nyquist frequency is around $2 \times 10^{-3}$ Hz. I think that the bounds in Fig. 9 are probably more relevant than those in Fig. 7.

**Reply**: You're correct. The entire paragraph has been re-written. And yes, you could say that the drop in resolved energy drops after $10^{-1}$, which is the reason we did not show the coherence curves much past that frequency.